# Rapid ice aggregation process revealed through triple-wavelength Doppler spectra radar analysis

Andrew I. Barrett[1,2], Christopher D. Westbrook[1], John C. Nicol[1], and Thorwald H. M. Stein[1]

[1]Department of Meteorology, University of Reading, Reading, RG6 6BB, UK
[2]Institute for Meteorology and Climate Research, Karlsruhe Institute of Technology, Karlsruhe, 76131, Germany

**Correspondence:** Andrew Barrett (andrew.barrett@kit.edu)

**Abstract.**

We have identified a region of an ice cloud where a sharp transition of dual-wavelength ratio occurs at a fixed-height for longer than 20 minutes. In this paper we provide evidence that rapid aggregation of ice particles occurred in this region creating large particles. This evidence comes from triple-wavelength Doppler spectra radar data that were fortuitously being collected. Through quantitative comparison of the Doppler spectra from the three radars we are able to estimate the ice particle size distribution (of particles larger than 0.75 mm) at different heights in the cloud. This allows us to investigate the evolution of the ice particle size distribution and determine whether the evolution is consistent with aggregation, riming or vapour deposition. The newly-developed method allows us to isolate the signal from the larger (non-Rayleigh scattering) particles in the distribution. Therefore, a particle size distribution retrieval is possible in areas of the cloud where the dual-wavelength ratio method would fail because the bulk dual-wavelength ratio value is too close to zero.

The ice particles grow rapidly from a maximum size of 0.75 mm to 5 mm while falling less than 500 m and in under 10 minutes. This rapid growth is shown to agree well with theoretical estimates of aggregation, with aggregation efficiency approximately 0.7, and is inconsistent with other growth processes, e.g. growth by vapour deposition or riming. The aggregation occurs in the middle of the cloud, and is not present throughout the entire lifetime of the cloud. However, the layer of rapid aggregation is very well defined, at a constant height, where the temperature is $-15\,°C$, and lasts for at least 20 minutes (approximate horizontal distance: 24 km). Immediately above this layer, the radar Doppler spectrum is bi-modal, which signals the formation of new small ice particles at that height. We suggest that these newly formed particles, at approximately $-15\,°C$, grow dendritic arms, enabling them to easily interlock and accelerate the aggregation process. The large estimated aggregation efficiency in this cloud is consistent with recent laboratory studies for dendrites at this temperature.

## 1   Introduction

Ice microphysical processes are an important part of cloud and precipitation formation; most surface precipitation begins as ice particles (Field and Heymsfield, 2015). However, numerical models, of either weather or climate, have difficulty accurately simulating ice cloud. For example, the CMIP5 models have regional cloud ice water paths that differ from observations by factors of 2–10 (Li et al., 2012). This challenge is partly because observations of ice particles are sparse and because processes

controlling the formation and evolution of ice particles, such as aggregation, are poorly understood and crudely parameterized in most models.

Additionally, measuring the number and size of ice particles within clouds is challenging. The two main methods, in-situ aircraft observations, and active remote sensing observations, both have their deficiencies. First, active remote sensing instruments, such as radar and lidar, are good at measuring the bulk scattering quantities, such as radar reflectivity. However, converting these bulk quantities to cloud microphysical properties requires numerous assumptions (e.g. the shape of individual hydrometeors, the particle size distribution). In contrast, aircraft observations measure the size and number of ice particles directly, but only within a small sample volume, at a single height at any given time, and only during sporadic case studies. Furthermore, ice particle size distributions have been shown to be biased as a result of shattering of ice particles on aircraft-mounted instrument inlets (Westbrook and Illingworth, 2009; Korolev et al., 2011), which results in an artificially increased concentration of small ice crystals.

Nevertheless, cloud microphysical observations and in particular particle size distributions are important for many applications. One important application is the better understanding of processes that occur within clouds. For example, size distributions measured from aircraft have been used to study aggregation in cirrus clouds (Field et al., 2006). Furthermore, the size distribution itself affects the relative importance of vapor deposition, riming and aggregation for ice particle growth. Vapor deposition and evaporation rates are proportional to first moment of particle-size distribution, while riming is related to higher moments (product of projected area and fall speed), while aggregation rates depend on the breadth of the particle-size distribution through the difference in fall speeds. So the shape and breadth of the particle-size distribution are an important control on the relative importance of the processes involved. Another important application is to provide observations with which numerical models can be evaluated and their parameterizations improved.

In this paper, we report of radar observations of one cloud system, where large vertical gradients in cloud microphysical properties were observed at a fixed height for at least 20 minutes. By exploring the radar data beyond the standard bulk quantities, and exploiting observations from multiple radars together with their Doppler spectra, we are able to estimate the size distribution of particles at different heights and therefore diagnose the most likely process for the rapid but consistent changes in cloud properties with height. The changes of cloud microphysical properties with height apparently result from rapid aggregation of ice particles. These observations were made using three co-located, vertically pointing radars at different frequencies (3, 35, 94 GHz).

Analysis of the radar Doppler spectra has previously been performed for the onset of drizzle in stratiform clouds (Kollias et al., 2011a, b) and the application of multi-frequency Doppler spectra has been used to determine the rain size distribution (Tridon and Battaglia, 2015; Tridon et al., 2017). For the ice phase, the three different frequencies have been used simultaneously to categorize rimed and unrimed particles from the surface (Kneifel et al., 2011, 2015, 2016) and from aircraft-based radar observations (Kulie et al., 2014; Leinonen et al., 2018; Chase et al., 2018). However, this is the first attempt to retrieve the ice particle size distributions from multi-frequency Doppler spectra observations. These retrievals are then used to evaluate the microphysical processes active within the clouds.

The aggregation process can be characterised by the aggregation kernel $k$ (Mitchell, 1988, eq. 9)

$$k = \frac{\pi}{4} E_{\mathrm{agg}} \left(D_1 + D_2\right)^2 \left|v(D_1) - v(D_2)\right| , \tag{1}$$

where $D_1$ and $D_2$ are the diameters of the two potentially-aggregating particles and $v(D)$ is the fall velocity of the particle. The aggregation efficiency of ice particles ($E_{\mathrm{agg}}$; the probability that two particles experiencing a "close approach" will collide and stick together) are typically low, although a large range of values have been reported and understanding of how aggregation efficiency varies with environmental parameters is still sparse. $E_{\mathrm{agg}}$ has previously been found to depend on both the particle habit and the temperature at which the collisions occur; however, a large range of values have been reported. An increase of the aggregation efficiency at about $-15\,°\mathrm{C}$ has been reported in several laboratory studies. One such study, (Hosler and Hallgren, 1960), where small particles were drawn past a large stationary ice target showed a weak temperature dependence of $E_{\mathrm{agg}}$ with a broad peak around $-12\,°\mathrm{C}$ and maximum values of $0.1$–$0.2$. Connolly et al. (2012) used a 10-m tall cloud chamber containing large concentrations of small ice particles settling under gravity and reported a much sharper peak of $E_{\mathrm{agg}}$ around $-15\,°\mathrm{C}$, with values of $0.4$–$0.9$, but the best estimate at other temperatures was below $0.2$. Keith and Saunders (1989) found aggregation efficiencies for planar snow crystals drawn past a cylindrical target of $0.3$–$0.85$ depending on the particle size. Hobbs et al. (1974) reported that both the maximum dimension of ice aggregates and the probability of seeing aggregates increased at around $-15\,°\mathrm{C}$, which was linked to the preferred formation of dendritic particles at this temperature. This is supported by other studies showing larger $E_{\mathrm{agg}}$ in the presence of dendritic particles. Mitchell et al. (2006) found $E_{\mathrm{agg}}$ of around 0.55 for clouds dominated by dendrites at cloud top, but much lower values around 0.07 when dendrites were not present. Low $E_{\mathrm{agg}}$ values of 0.09 were also found for tropical anvil clouds where dendritic particles were not present at temperatures of $-3\,°\mathrm{C}$ to $-11\,°\mathrm{C}$ (Field et al., 2006). In the early stage of aggregation, Moisseev et al. (2015) reported that the aggregates were made up of a small number of dendritic particles. These studies seem to suggest that dendrites, which typically form at around $-15\,°\mathrm{C}$, can significantly increase the aggregation efficiency because the dendritic branches interlock with other particles, whereas the aggregation efficiency is much lower when dendritic particles are not present. In this study, retrievals from radar observations will be used to estimate the aggregation efficiency and will be compared with the laboratory-derived values.

Barrett et al. (2017) showed that the assumed particle size distribution is the single-largest sensitivity in the model physics for mixed-phase altocumulus clouds. The importance of correctly simulating the ice particle size distribution has been shown in several other studies (Pinto, 1998; Harrington et al., 1999; Field et al., 2005; Morrison and Pinto, 2006; Solomon et al., 2009). Therefore understanding and correctly implementing the aggregation process in numerical models of cloud physics is important for the overall development of the cloud system.

This paper is organised with an overview of the instruments and data in section 2, an overview of the case study in section 3 and details about the retrieval in section 4. Section 5 details the cloud properties retrieved and their uncertainties and section 6 summarizes the evidence for aggregation, with conclusions drawn in section 7.

## 2  Data and Methods

We use data from three co-located radars at the Chilbolton Observatory in Hampshire, Southern England on the afternoon of 17 April 2014. The radars operate at frequencies of 3 GHz (9.75-cm wavelength, 25-m antenna, 0.28° beamwidth; Goddard et al., 1994a), 35 GHz (8.58-mm wavelength, 2.4-m antenna, 0.25° beamwidth; Illingworth et al., 2007) and 94 GHz (3.19-mm wavelength, 0.46-m antenna, 0.5° beamwidth; Eastment, 1999). The 35- and 94-GHz cloud radars are situated immediately next to one another, whereas the 3-GHz radar is sited less than 50 m away (Fig. 2). The sampling of the three radars was synchronized to within 0.1 seconds and full pulse-to-pulse power and phase measurements were recorded. For the 3-GHz radar, Doppler spectra were calculated every second and incoherently averaged over 10 seconds. For the 35-GHz and 94-GHz cloud radars, spectra were calculated every 0.11 and 0.08 seconds respectively and again incoherently averaged over 10 seconds. Assuming typical wind speeds of 20 m s$^{-1}$ aloft, the averaged spectra correspond to a 200-m section of cloud. Ground clutter was removed from the spectra by masking returns with velocity near zero. Noise levels were estimated from measurements beyond the range of meteorological echoes (> 10 km) and subtracted from the individual spectra prior to averaging. The data from each radar was interpolated on to common range and velocity grids (60-m range by 0.0195 m s$^{-1}$ velocity).

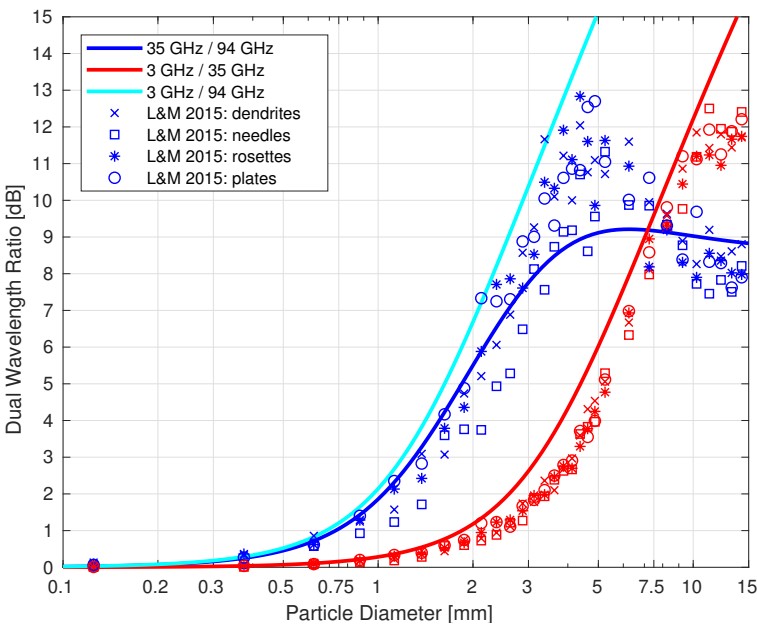

**Figure 1.** Dual-wavelength ratios as a function of ice particle diameter for the three pairs of radar frequencies used in this study. Dual-wavelength ratios from the Westbrook et al. (2006, 2008a) scattering model are shown with solid lines. For comparison, mean dual-wavelength ratios of unrimed aggregates within 250-micron-wide diameter bins from Leinonen and Moisseev (2015) are shown by points for four different aggregate-monomer types.

Because of the large antenna, it is necessary to apply a near-field correction to the 3-GHz data at heights below about 6 km (Sekelsky, 2002). This correction factor was derived empirically by comparing 3-GHz reflectivity profiles against those

**Table 1.** A summary of the terminology used throughout this paper, where F denotes the radar frequency.

| symbol | variable name | variable definition | unit |
|---|---|---|---|
| $Z_F$ | radar reflectivity | total radar cross sectional area of scatterers within the target volume | dBZ [$Z = mm^6\ m^{-3}$] |
| $DWR_{F1/F2}$ | dual-wavelength ratio | $Z_{F1} - Z_{F2}$ | dB |
| $sZ_F$ | spectral reflectivity | radar reflectivity per Doppler-spectra velocity bin | dBX [$X = mm^6\ m^{-3}\ (m\ s^{-1})^{-1}$] |
| $sDWR_{F1/F2}$ | spectral dual-wavelength ratio | $sZ_{F1} - sZ_{F2}$ | dB |

measured by the 35-GHz instrument (which has a much smaller antenna) in a number of Rayleigh scattering ice clouds. The magnitude of the correction was 1 dB at 5 km, rising to 3 dB at 3 km.

## 2.1 Data quality, calibration and attenuation correction

To account for potentially imperfect calibration and attenuation by atmospheric gases and liquid water in the lower troposphere,
the 35- and 94-GHz reflectivity is corrected relative to 3-GHz radar. The 3-GHz radar is absolutely calibrated to within 0.5 dB, using the method of Goddard et al. (1994b). The radar reflectivity value from the cloud radars (35 and 94 GHz) was adjusted to match the 3-GHz radar reflectivity in each profile so as to remove any calibration or attenuation offsets. The adjustment amount was estimated in regions where Rayleigh scattering was expected at all three wavelengths[1] and hence where the reflectivity should be the same from each radar. The adjustments reduce the median difference in reflectivity ($Z$) in the Rayleigh scattering
areas to 0 dB. The same adjustment to $Z$ (in dB) is made throughout the profile. A different correction is applied individually to each 10-second profile; the equivalent dB correction is also applied to the Doppler spectra power within each profile. This adjustment works well because the majority of the attenuation by atmospheric gases and liquid water occurs below cloud base. In other cases, where cloud base is lower or with embedded liquid water layers, a different treatment would be necessary.

The multi-wavelength approach allows us to measure the diameter of ice particles that are comparable in size to the shortest
radar wavelength, or larger (e.g. Kneifel et al., 2015, 2016). For ice particles comparable in size to the radar wavelength, non-Rayleigh scattering becomes important. For suitably large particles, it becomes possible to size the particles based on the different radar returns at different wavelengths.

In contrast to the bulk retrieval that makes a single retrieval for particles of all fall velocities together, the Doppler spectra approach allows for retrievals of particle size and number concentration to be made separately on particles of distinct fall
velocities. We can use the multi-wavelength approach to determine the representative particle size from the "spectral dual-

---

[1]Based on an analysis of reflectivity differences, Rayleigh scattering is assumed where the 3-GHz reflectivity is below 5 dBZ and the absolute difference between the 3- and 94-GHz velocity measurements is less than 0.025 m s$^{-1}$. Measurements were also excluded where the 3-GHz reflectivity was less than $-10$ dBZ to avoid effects of residual ground clutter.

wavelength ratio" (sDWR; i.e. the difference in reflectivity of particles within a small range of fall velocities; see Table 1 for a full summary of radar quantities used in the paper), but can additionally separate the particles based on their fall velocity allowing us to retrieve the ice particle size distribution.

A correction to the velocities measured by the radar is also applied. Unfortunately, the three radars were not precisely vertically pointing for this case (as determined by biases in the mean Doppler velocity in the Rayleigh-scattering part of the cloud) and initial testing suggested that there was a large sensitivity to the velocity offsets in the spectra (see section 6.1). The 3-GHz radar was pointing vertically, but after analyzing the data, the 35- and 94-GHz radars were determined to be off-zenith by approximately $0.2°$ and $0.15°$ respectively in opposing directions. These offsets were determined by assessing the mean Doppler-velocity differences between the three radars as a function of height. The correlation of these velocity differences with the atmospheric wind profile (determined from ECMWF forecast fields) enabled an estimation of the pointing angle errors.

The mispointing of the radars is small and likely does not result in a substantial mismatch in sample volume given the 10-second integration time. However, this small mispointing means that the radar detects a small component of the horizontal wind in addition to the fall velocities of the ice particles. Although the pointing angle error is small, the horizontal wind component detected is of the order of a few centimeters per second, which is sufficiently large to affect our comparison of the Doppler spectra from the three radars. Therefore, we have made a correction to the velocity measurements for the 35- and 94-GHz radars to ensure that the spectra are well aligned and can be compared. This correction is important because even a small shift in velocity can substantially affect the estimates of sDWR. In practice, the correction applied is $+0.0585\,\mathrm{m\,s^{-1}}$ (3 velocity bins) for the 35-GHz radar and $-0.0390\,\mathrm{m\,s^{-1}}$ (2 velocity bins) for the 94-GHz radar throughout the cloud layer. This correction is imperfect; however, we do not have independent measurements of the horizontal wind speed with sufficient accuracy and high enough vertical resolution to make a reliable height-dependent correction or indeed any direct measurement of the mispointing. Radiosonde data and ECMWF model output shows that the horizontal wind speed was near-constant with height throughout the cloud layer on this day, and inspection of many individual Doppler spectra indicate that our simple correction aligns the spectra very well in this case.

To reduce the noise in the spectra, each individual spectrum has been smoothed in velocity space by averaging over a $0.0585$ $\mathrm{m\,s^{-1}}$ window, which equates to three velocity bins. We mask out regions where significant turbulence is present because the vertical air motions are large and vary on small time and space scales compared to the particle fall velocities that we are trying to measure. Near the cloud base, there is a layer of substantial turbulence caused by sublimation of ice particles as they fall into subsaturated air and this leads to destabilisation of the atmosphere in this layer. In this turbulent layer, the implicit assumption that measurements at a specific velocity are of a single particle size is invalid. Hence, we identify regions where turbulence is altering the spectra, by calculating the contribution of turbulence to spectral width using O'Connor et al. (2005, eqns. 10–15). Points where the velocity variance from turbulence exceed a threshold value of $10^{-3}\,\mathrm{m^2\,s^{-2}}$ are not considered when performing our retrievals. This threshold value was chosen such that all affected regions were suitably masked and in the remaining data that the width of the Doppler spectrum was determined by microphysical rather than turbulent contributions. Additionally, any points in the spectra that are 20 dB down from the peak of the spectra are removed in order to minimise the impact of noise.

## 3 The case - 17 April 2014

Figure 3a shows the radar reflectivity measured at Chilbolton for the thick stratiform ice cloud observed on 17 April 2014. This cloud formed in north-westerly flow, ahead of a cold front. The surface cold front reached Chilbolton at about 1800 UTC. The front was not associated with any surface precipitation at Chilbolton, and only very light precipitation across some other parts of Southern England.

The evolution of the cloud reflectivity and the ratio of 35-GHz and 94-GHz reflectivity are shown in Figure 3. The cloud top height was approximately 9 km, where 35-GHz reflectivity values are around $-15$ dBZ, and increase to 19 dB at approximately 4 km altitude, near cloud base. The temperature at cloud top was $-45$ °C and the freezing level was at about 2.7 km. Throughout most of the cloud the DWR values are below 1 dB. However, at around 4 km altitude there is a rapid increase of DWR with decreasing height which indicates an increase in particle size such that the backscattered return at 94 GHz is no longer from Rayleigh scattering. The region of these large DWR values is consistent in height (onset at 4.5 km altitude; Fig. 3c), and is evident for at least 35 minutes. The largest DWR values occur at around 1615 UTC, with peak values reaching 7 dB. The profile of DWR values at 1615 UTC is shown in Fig. 3c.

Radar data from a larger portion of the same cloud was analysed in Stein et al. (2015). Earlier in the day (before 1540 UTC), the cloud did not show this sharp transition to high DWR values around 4.5 km. Stein et al. (2015) also used the triple-frequency radar data to determine that the cloud contained primarily aggregate snowflakes, consistent with the Westbrook et al. (2006, 2008a) scattering model (lines in Fig. 1). Scattering properties of unrimed aggregates from Leinonen and Moisseev (2015) are also consistent with observations, and give very similar characteristics to the Westbrook et al. (2006, 2008a) scattering model (points in Fig. 1). We focus on the time from 1545 to 1620 UTC, where there are dual wavelength ratios up to 8 dB below 4.5 km (Fig. 3b,c).

We attempt to understand what causes the rapid change in cloud properties during this period of substantial DWR$_{35/94}$ and the rapid change in height. Looking at the spectral reflectivity at each height ($sZ_{35}$; Fig. 4a) together with the spectral dual wavelength ratio ($sDWR_{35/94}$; Fig. 4b) reveals the changes of the cloud properties with altitude. From these data, the origin of the large changes in the sharp transition can be identified. At 5.4 km, there is an increase in the signal coming from slow-falling particles (0.4–0.6 ms$^{-1}$; Fig. 4a). At this height, only the fastest falling particles have sDWR$_{35/94}$ > 1dB. At 4.5 km, the reflectivity and spectral reflectivity of the slow-falling particles has increased. The sDWR$_{35/94}$ increases up to 8 dB for the fastest-falling particles, and by 4 km the increase in sDWR$_{35/94}$ is seen for the majority of particles. Interestingly, the fall velocity of these particles does not increase as the particles grow larger and produce large sDWR$_{35/94}$ values.

## 4 Retrieval of the ice particle size distribution

To retrieve the ice particle size distribution from the cross-calibrated and velocity-matched Doppler spectra (see section 2.1) at three wavelengths, we use the method described below. The method is illustrated at three separate heights in Figure 5. The following is calculated for each individual velocity bin, within each radar range gate and at all times:

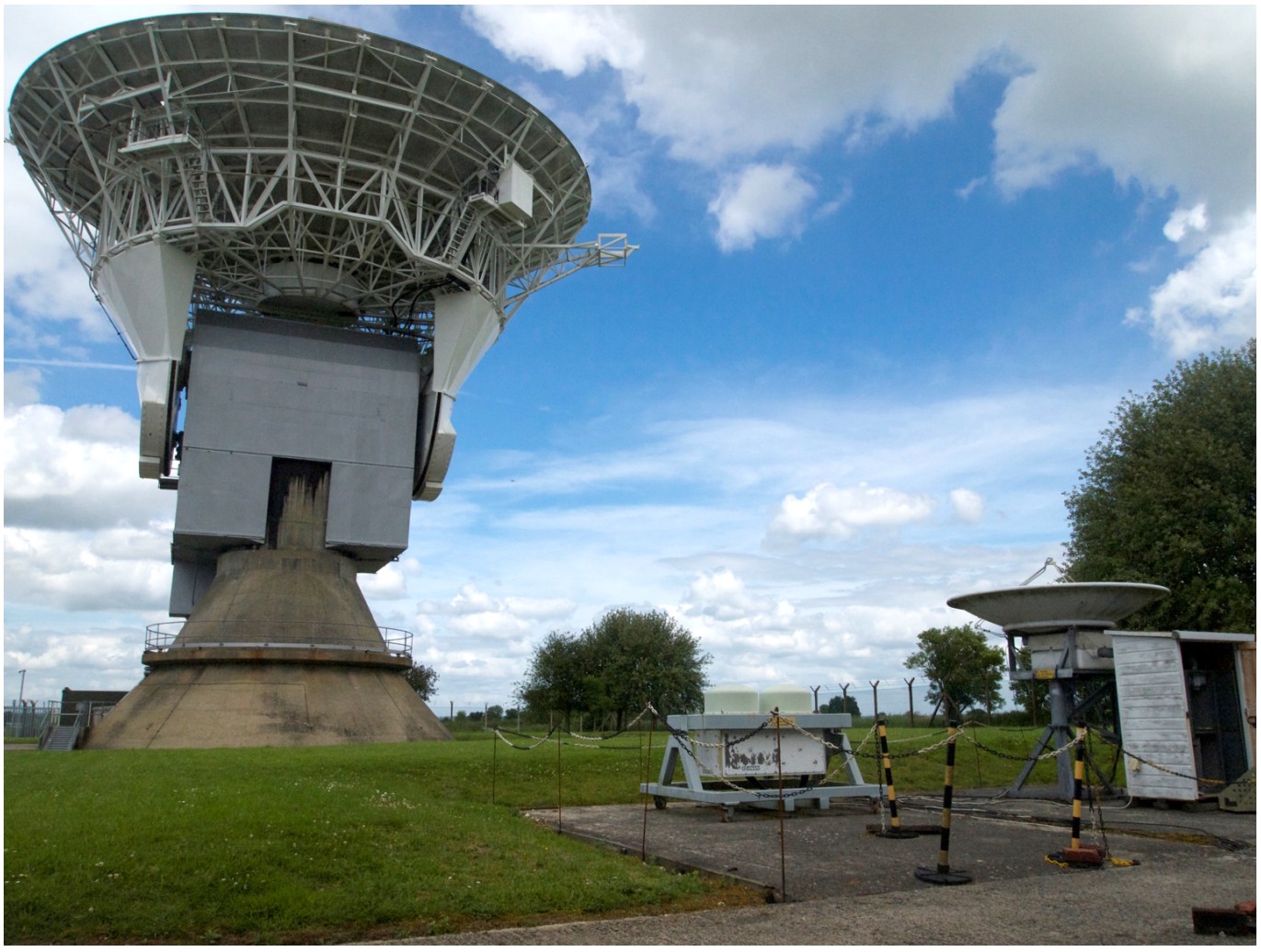

**Figure 2.** A photograph of the three co-located radars at the Chilbolton Observatory, Hampshire, England. From left to right: the 3-GHz CAMRa radar, 94-GHz radar and 35-GHz radar.

1. Calculate the spectral dual-wavelength ratio ($sDWR = sZ_{35} - sZ_{94}$). This is simply calculated as the difference between the spectral reflectivity ($sZ$) at 35 and 94 GHz (Fig. 5a,d,g).

2. Determine the particle diameter $D$ from $sDWR$. The relationship between particle diameter and particle DWR from the Westbrook et al. (2006, 2008a) scattering model (Fig. 1) is used to convert the $sDWR$ value to particle diameter. We use this scattering model based on its good agreement with observational data for this case (Stein et al., 2015); other scattering models may be more appropriate for different cases.

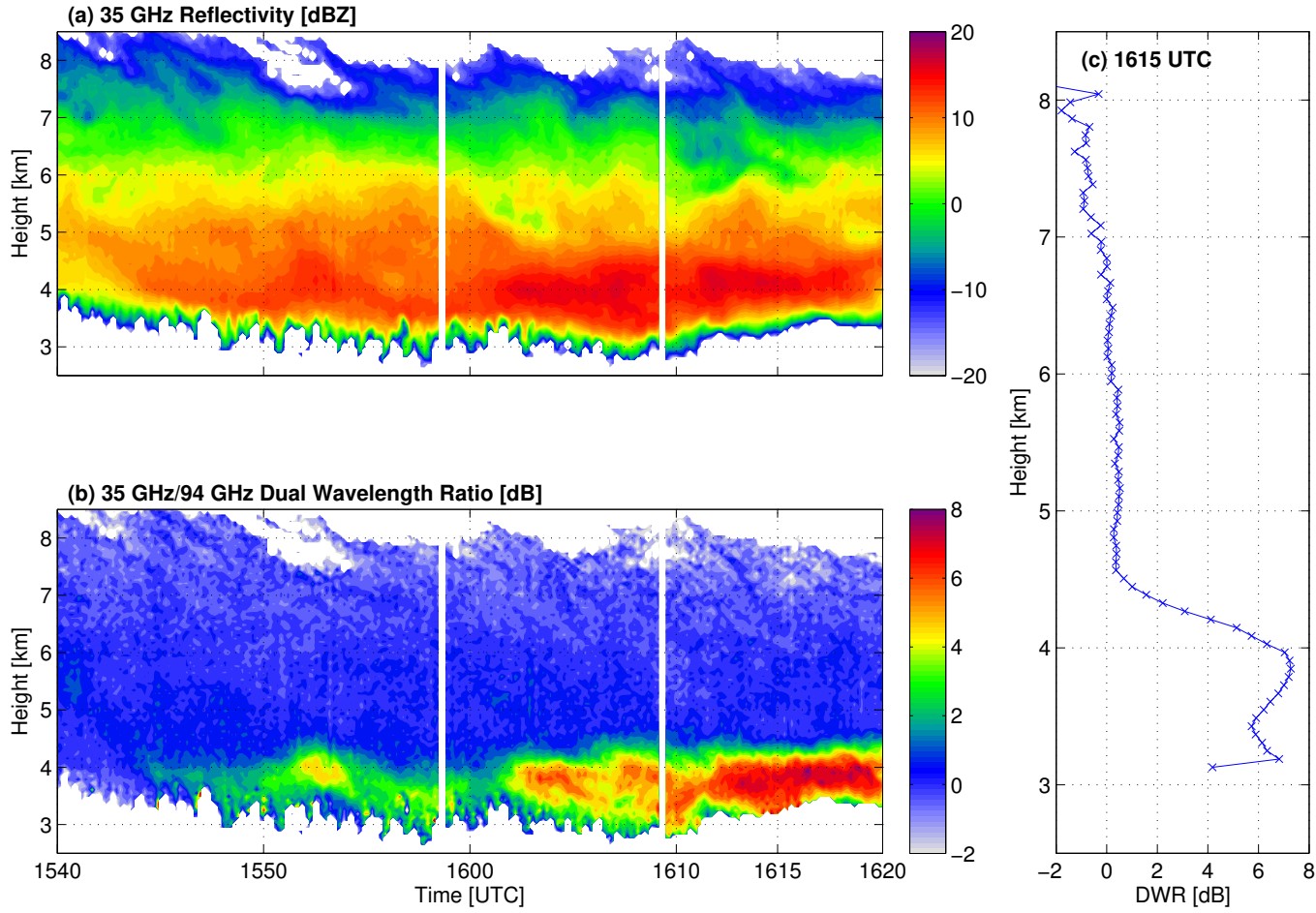

**Figure 3.** Overview of the cloud structure on 17 April 2014 showing the a) 35-GHz radar reflectivity and b) the ratio of 35-GHz reflectivity to 94-GHz reflectivity throughout the sampling period. c) The vertical profile of DWR at 1615 UTC.

3. Calculate the mass $m$ of an ice particle with diameter $D$, assuming the Brown and Francis (1995) mass-size relationship of $m = 0.0185D^{1.9}$ for all ice particles. Use of this mass-diameter relationship is supported by (Stein et al., 2015), who found that the fractal dimension of snowflakes on this day was $1.9$ and hence the exponent of $1.9$ is appropriate; other mass-diameter relationships may be more appropriate for different cases.

4. Determine the radar reflectivity of a single ice particle with diameter $D$ and mass $m$ using the scattering model.

5. Determine the total number of particles within the velocity bin. This is calculated by dividing the total spectral reflectivity $sZ$ by the single-particle reflectivity calculated in the previous step.

The size and number of ice particles within the velocity bin is now known. The particle size distribution can be estimated by performing this same process for each velocity bin.

Up to this point, we have determined the diameter $D$ of the ice particles within each velocity bin, and also the particle size distribution d$N$/d$V$ (where d$N$ is the concentration of ice particles with velocity between $V$ and $V$+d$V$). We can convert d$N$/d$V$ to the ice particle size distribution d$N$/d$D$ (concentration of ice particles with diameter between $D$ and $D$+d$D$). This is the common way to express a particle size distribution that is independent of the measuring sample interval (d$D$ or d$V$).

To do so, we need to know the relationship between the velocity bin width d$V$ and the diameter bin width d$D$. To determine this, we use a 300-m by 90-s window (5 range gates by 9 individual averaged spectra) centered on the current radar pixel and compute the power-law fit to the measured Doppler velocity and retrieved diameter values, of the form $V = cD^d$. A power-law relationship is used because it is both easily differentiable and common in microphysical scaling relationships (e.g. Locatelli and Hobbs, 1974). We use the differential of this power-law fit to compute $\frac{dV}{dD}$ - the diameter bin width for each velocity bin.

The size distribution is then calculated as

$$\frac{\mathrm{d}N}{\mathrm{d}D} = \frac{\mathrm{d}N}{\mathrm{d}V}\frac{\mathrm{d}V}{\mathrm{d}D} \quad . \tag{2}$$

There is a relatively large sensitivity of the retrieved size distribution to the power-law fit, but this is primarily in terms of the number concentration, rather than the diameter of the particles or the shape of the size distribution (see section 6.1 for a complete sensitivity analysis).

Retrieval of the size and number concentration of ice particles is only possible for particles larger than about 0.75 mm in diameter (corresponding to a sDWR$_{35/94}$ of about 1 dB, Fig. 1). For smaller particles, $sZ$ is very similar at all three radar frequencies and differences are not easily distinguished from noise in the spectra. For particles larger than about 3 mm diameter, the sDWR$_{35/94}$ saturates at about 8–9 dB (Fig. 1) as a result of the fractal geometry of the aggregates (see Stein et al., 2015) and therefore retrieval of particle diameter from $s\mathrm{DWR} - 35/94$ is no longer possible. Therefore, where sDWR$_{35/94}$ is larger

than 6 dB, the diameter and number concentration are retrieved using sDWR$_{3/35}$ instead, following the same method as above. This pair of frequencies does not saturate until significantly larger particle diameters and therefore, for larger particles, has a larger sensitivity to change in diameter than for the 35/94 GHz pair. We do not use the 3/35 GHz pair for the full range of particle diameters because the 3-GHz is affected more by noise than the 35-GHz spectra, and therefore negatively impacts the retrieval of particle sizes when DWR is small. It would be equally valid to calculate the size and number concentration of the

larger particles using the 3/94-GHz pair instead, and this indeed enables a consistency check that the retrieval works well and that the input Doppler spectra are well matched.

## 5   Retrieved cloud properties and validation

Throughout most of the cloud, the 35/94-GHz dual-wavelength ratio (DWR$_{35/94}$) is near zero ($<$1 dB) (Fig 3b), implying that the ice particles are relatively small and are still in the Rayleigh scattering regime at 94 GHz (maximum diameter 0.75 mm).

DWR only exceeds 2 dB after 1545 UTC and between 4.3 km and cloud base.

From 1600 to 1620 UTC, there is a sharp transition from DWR$_{35/94}$ $<$ 1 dB at 4.5 km to peak DWR$_{35/94}$ values at 4 km, with the maximum DWR$_{35/94}$ = 8 dB. The altitude of this sharp transition is consistent after 1602 UTC, with the largest DWR$_{35/94}$ values being present after 1610 UTC. There is also evidence of this transition layer as early as 1545 UTC.

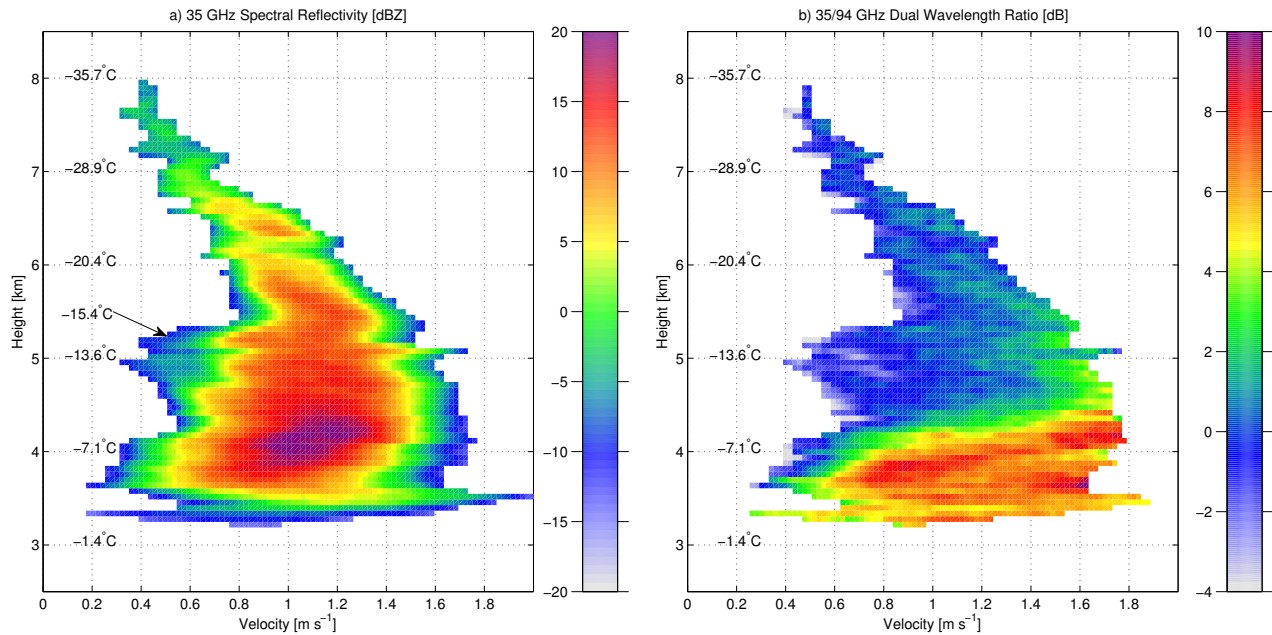

**Figure 4.** Height profile of a) spectral reflectivity at 35 GHz ($sZ_{35}$) and b) spectral dual wavelength ratio (sDWR$_{35/94}$) recorded at 1615 UTC. Temperatures from the ECMWF model at 1600 UTC are shown every 1 km and at 5.3 km where the small-particle mode is first evident.

More detail can be seen by examining the Doppler spectra for the different radars at a few fixed heights in detail. The Doppler spectra measured at 5.89, 4.81 and 4.15 km (Fig. 5a,d,g) show three spectra with rather different shapes. At 4.15 km, the spectra has only a single mode but throughout most of the velocity range $sZ_{35}$ is much greater than $sZ_{94}$. The sDWR$_{35/94}$ reaches 8 dB (Fig. 5h) and the largest particles are sized at around 5 mm. The retrieved size distribution is approximately inverse exponential (Fig. 5i).

At 5.89 km (top row of Fig. 5), in contrast, the spectra for all three radars are very similar with a single peak; all sDWR$_{35/94}$ values below 1 dB (Fig. 5b). The small sDWR$_{35/94}$ values mean that it is not possible to reliably size the ice particles here, other than to say that they are all smaller than 0.75 mm.

About 1 km lower in the cloud, at 4.81 km (second row of Fig. 5), the mean velocity and reflectivity have both increased, but there is also a bi-modal structure to the spectra captured at both frequencies. This second mode is related to newly formed, small ice particles that are falling slower than the majority of older, larger ice particles. Furthermore, at 4.81 km, there are larger and faster-falling particles present than at 5.89 km. The largest sDWR$_{35/94}$ values now approach 4 dB (Fig. 5e), and particles larger than 0.75 mm are present, with the largest retrieved diameter of 1.2 mm. The size distribution (Fig. 5f) of the reliably-sized particles (those larger than 0.75 mm and outside the gray region of the plot) is inverse exponential.

The consistent and narrow range of heights over which this rapid change in size occurs is just below the region where new particles are seen around 5.4 km and the Doppler spectra is bi-modal (Fig. 5d). These new particles fall slowly, which suggests

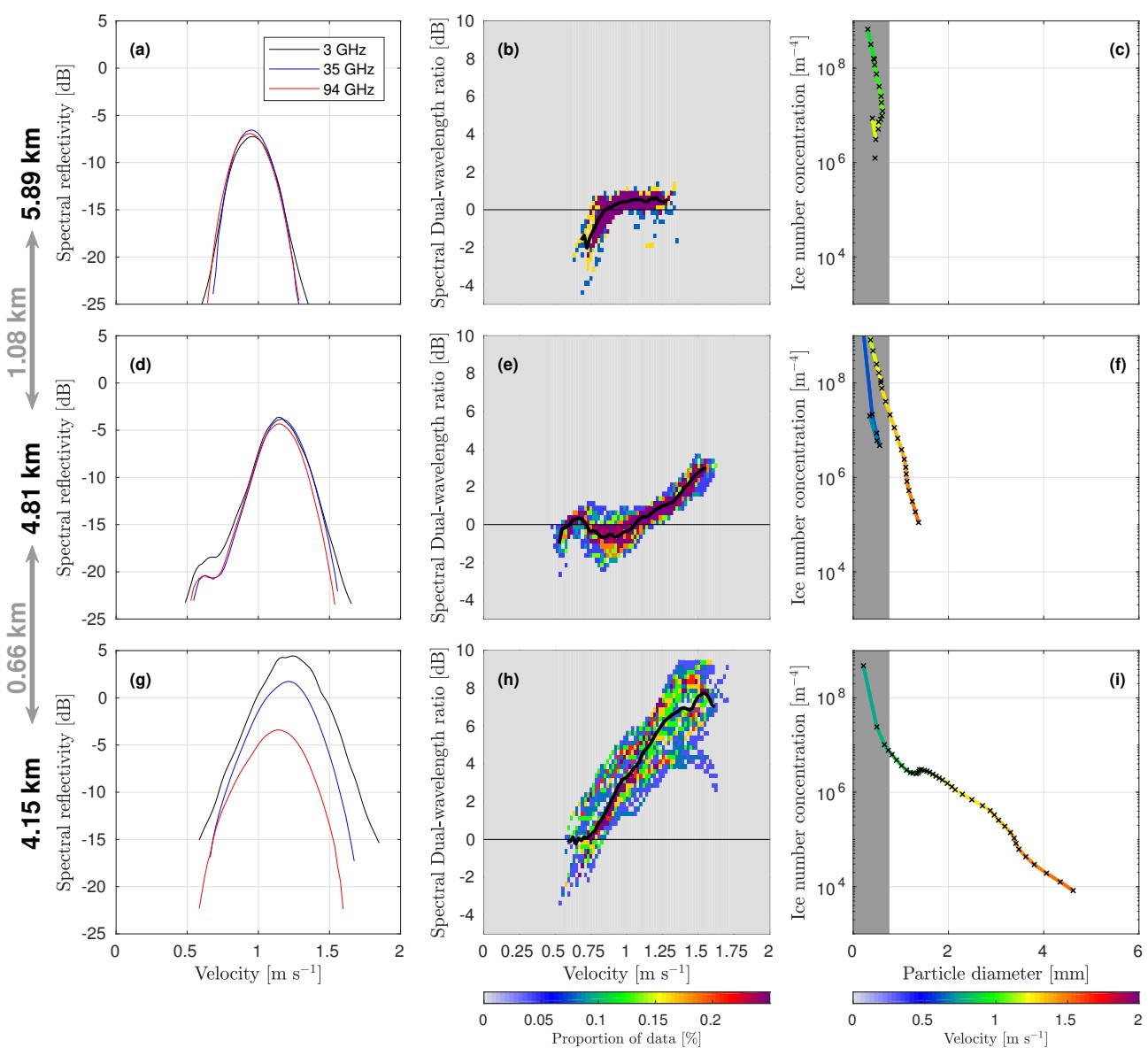

**Figure 5.** Illustration of the retrieval method and the retrieved size distribution at three heights at 1615 UTC. a-c are just above the layer of secondary ice nucleation, d-f are within that layer and g-i are below this layer, where the dual-wavelength differences are largest. a,d,g show the 3-, 35- and 94-GHz spectra at that height. b,e,h show the distribution of $sDWR_{35/94}$ data points within a window around the central time (90-s by 300-m), with the black line denoting the median power difference for each velocity bin. c,f,i show the retrieved ice particle size distribution, with the color of the line relates to the velocity of the data used to determine that data point. The gray shaded region marks particle diameters smaller than 0.75 mm, where there is no reliable information available to size the ice particles. The higher altitude plots are from earlier times to account for an approximately 1 m s$^{-1}$ fall velocity of the ice particles.

that they are small and are formed at this level. These particles begin to fall faster as they grow in size. Particles forming around $-15$ °C would initially grow as dendrites (Takahashi et al., 1991). As these particles grow, the $\text{sDWR}_{35/94}$ starts to increase for the larger (faster falling) particles, which we take to be aggregates. This increase in $\text{sDWR}_{35/94}$ implies an acceleration of the aggregation process at this height.

The reduction of the size distribution slope between 4.81 km and 4.15 km remains consistent for at least 30 minutes from 1545 UTC onwards, but is not present earlier in the cloud. The observations shown in Fig. 5 are similar throughout this time period, which explains the sharp increase of DWR between 4.8 and 4.1 km (Fig. 3) during this time period.

## 5.1  Evolution and validation of retrieved size distributions

To evaluate how accurate the retrieved ice particle size distributions are, we would ideally like to compare against in-situ
data. However, in-situ observations are not available for this case. Therefore, we evaluate the retrievals against other retrieval methods.

By fitting an inverse-exponential to the retrieved particle size distribution data from our Doppler spectra method, we can estimate the slope of the size distribution, $\Lambda$ in $\mathrm{d}N/\mathrm{d}D = N_0\exp(-\Lambda D)$ (Fig. 6a). By means of verification, we also calculate the slope of a purely inverse-exponential size distribution fitted to match the $\text{DWR}_{35/94}$ values only (Fig. 6b). There is excellent
agreement between the two methods in the regions where the size distribution is broader and less steep. Fig. 6c shows a 2-minute average of $\Lambda$, which again shows the excellent agreement throughout the whole profile, particularly the height of the rapid change of $\Lambda$ between 5 and 4 km. The only region of disagreement is just below 4 km, where the spectra method suggests even broader size distribution than the DWR method. This could be evidence that the inverse-exponential size distribution approximation in this region is not appropriate or because $\text{DWR}_35/94$ has almost saturated at 8–9 dB. However, both methods
agree that there is a rapid increase in ice particle size occurring as they fall from 4.5 km to 3.6 km and a broadening of the ice particle size distribution. In the next section, we present evidence that this rapid change is occurring as a result of aggregation and not occurring through vapor deposition or riming.

The spectral method developed here is more sensitive to the presence of a few large particles than the DWR method. With the spectral method, the influence of a few non-Rayleigh scatterers can be seen in the spectra before the reflectivity of the
individual scatterers is large enough to contribute significantly to the total reflectivity (which is a weighted average of sDWR over all particles). Therefore, the retrieved particle size distributions higher in the cloud are more reliable with the spectral method than the DWR method, because we are able to isolate the signal from the larger particles in the distribution. However, the spectral method is sensitive to noise in the spectra, and hence when the overall signal becomes weak, and the noise is therefore a more significant contributor, the retrieved particle size distributions are also noisy.

## 30  6  Evidence for rapid aggregation of dendrites

In this section we examine whether the changes in particle size and size distribution could be explained by processes other than aggregation. Specifically we address whether vapor deposition or riming could lead to the observed changes.

Ice particles grow from smaller than 0.75 mm in diameter (DWR<1 dB), above this transition layer, to larger than 5 mm by the time they reach 4 km (Figure 3c). Mean radar Doppler velocities just above this transition layer are 1–1.2 m s$^{-1}$ (Figure 5d), indicating that on average ice particles will take 400–500 seconds to fall from 4.5 to 4 km, although the largest particles responsible for the large DWR values will fall faster than the average particle.

The growth of ice particles by vapor deposition cannot produce large ice particles sufficiently quickly to match our observations. Calculations using the vapor deposition growth equation from Pruppacher and Klett (1978) are presented to demonstrate this. The equations used were,

$$\frac{\mathrm{d}m}{\mathrm{d}t} = \frac{4\pi\, C\, SS_i\, F}{\left(\frac{L_s}{R_v T} - 1\right)\frac{L_s}{KT} + \frac{R_v T}{e_{si}(T)D}} \quad , \tag{3}$$

$m = 0.0185 D^{1.9} \quad ,$                                                                     (4)

where the rate of change of particle mass $m$ with time $t$ is a function of the ice particle capacitance $C$ (assumed $= D/4$ here, following Westbrook et al. (2008b), where $D$ is the diameter), supersaturation with respect to ice $SS_i$ and ventilation coefficient $F = 0.65 + 0.44 * 0.6^0.33 \mathrm{Re}^0.5$. Re is the Reynolds number; $\mathrm{Re} = \rho D V(D)/\mu$, calculated from the air density $\rho$, particle diameter $D$, terminal velocity $V(D)$ and dynamic viscosity of air $\mu$. Terms on the denominator are the latent heat

of sublimation $L_s$, the specific gas constant for vapor $R_v$, temperature $T$, thermal conductivity of air $K$, and saturated vapor pressure over ice $e_{si}$, (4) is the Brown and Francis (1995) mass-size relationship.

     These calculations, for a liquid-saturated atmosphere at $-10$ °C, show that typical ice particles would, at their absolute fastest, take over 40 minutes (2534 seconds) to grow from 0.75 mm to 5 mm in diameter. Similarly, Fukuta and Takahashi (1999) calculate that it takes over 30 minutes to grow a particle of 3 mm through vapor deposition. We therefore can rule out

pure vapor deposition as the source of the largest particles, which develop in less than 10 minutes.

     Riming of the ice particles by collecting liquid water is another possible explanation; however, there is no evidence of significant supercooled liquid water present at this height. There were no strong backscatter returns in the lidar measurements (not shown) which would indicate the presence of liquid droplets, and the liquid water path measured by the microwave radiometer is below the noise level of the instrument (about 20 g m$^{-2}$) throughout the observation period. Furthermore, the

triple-frequency analysis for the scattering models in Stein et al. (2015) do not show agreement with the expected triple-wavelength signature of rimed particles (Kneifel et al., 2016), but rather for aggregate snow crystals.

     The sharp and consistent transition of cloud properties with height after 1545 UTC is therefore likely a result of aggregation. The first indication that aggregation is the most important process in this part of the cloud is the continual decrease of $\Lambda$ (the slope of the ice particle size distribution) with height down from the top of the transition layer. This change with height

indicates that there are more large particles and fewer small particles as the particle size distribution evolves, consistent with aggregation.

$$\frac{d\Lambda}{dz} = \frac{\Lambda}{b\chi_f}\frac{d\chi_f}{dz}\left[1 - \frac{2\Gamma(b+\delta+1)\Gamma(b+d+1)}{\Gamma(\delta+1)\Gamma(2b+d+1)}\right] - \frac{\pi E_{\mathrm{agg}}I_l\chi_f\Lambda^{b+d-1}}{4abc\Gamma(b+d+1)\Gamma(2b+d+1)} \tag{5}$$

We calculated the expected change of $\Lambda$ with height using (5), following Mitchell (1988), for several different values of aggregation efficiency ($E_{\text{agg}}$). In (5), $a$, $b$ are constants in the mass-diameter relationship $m = aD^b$; $c$, $d$ are constants from the fall velocity-diameter relationship $V = cD^d$; $\delta = 1.0$ following Mitchell (1988); $\Gamma$ is the gamma function; $\chi_f$ is the snow-flux in kg m$^{-2}$ s$^{-1}$ and $I_l$ is calculated from eqn. 20 of Mitchell (1988), dependent on $b$ and $d$ - in our calculations it takes the value of 11.524. These calculations assume that aggregation and vapor deposition together are the primary processes affecting the evolution of the size distribution, and that changes to the total mass are only due to vapor deposition or sublimation not the accretion of liquid drops.

To estimate the aggregation efficiency in this part of the cloud, we need to know the slope of the particle size distribution at the top of the layer and the vertical profile of snow flux. The $\Lambda$ value is estimated from the retrieved size distribution. The snow flux profile is estimated from the retrieved particle diameters, converted to a mass, multiplied by the measured Doppler velocity and then integrated across the spectra. Using the retrieved profile of size distribution properties and snow flux profile at 1615 UTC as input, the expected change of $\Lambda$ with height for $E_{\text{agg}}$ values of 0.2, 0.7 and 1.0 are shown in Fig. 6c. The evolution of $\Lambda$ between 4.5 and 4.0 km altitude, as calculated by either the Doppler spectra method or the simpler DWR method, are both consistent with theoretical evolution with $E_{\text{agg}}$ around 0.7. The $E_{\text{agg}} = 0.2$, reported in Hosler and Hallgren (1960) cannot reproduce the observed broadening of the size distribution through this shallow layer of cloud, and leads to $\Lambda$ being overestimated by almost an order of magnitude at 3.5 km. $E_{\text{agg}} = 0.7$ is at the higher end of values reported in the literature. However, Connolly et al. (2012) found $0.4 < E_{\text{agg}} < 0.9$ at $-15$ °C, whereas for all other temperatures sampled the best estimate was $E_{\text{agg}} \leq 0.2$. Similarly, Field et al. (2006) reported that $E_{\text{agg}}$ values greater than unity were required for small particles for a good fit to observed aggregation within tropical cirrus anvils. Our results are consistent with the high values of $E_{\text{agg}}$ of Connolly et al. (2012) at $-15$ °C, but do not support the $E_{\text{agg}} < 0.2$ reported by Hosler and Hallgren (1960). This suggests that the free-fall experiments in the 10-m cloud chamber may be more representative of the natural aggregation in the atmosphere than the stationary target experiments of Hosler and Hallgren (1960). Connolly et al. (2012) speculate that the higher $E_{\text{agg}}$ at $-15$ °C is because the dendritic branches of the crystals are able to interlock and that this can increase $E_{\text{agg}}$ by at least a factor of 3. Increased aggregation efficiency in the presence of dendritic crystals also agrees with observations by Hobbs et al. (1974). Our observations are consistent with these hypotheses.

Further evidence to support the hypothesis of rapid aggregation in this part of the cloud is seen in the vertical profiles of Snow Flux and Number Flux (Fig. 7). These quantities have been calculated by determining the number and total mass of ice particles at each height, and for each velocity bin from the Doppler spectra retrieval. The mass (or number) in each velocity bin is then multiplied by the Doppler velocity measured by the radars in order to determine the flux. Only flux values of particles >0.75 mm in diameter are shown, because the number of smaller particles cannot be reliably estimated with this combination of radars. Confidence is given to the reliability of our retrievals by the coherent structures seen in time and height (Fig. 7a,b). The vertical profile of Snow Flux and Number Flux (Fig. 7c) also supports our rapid aggregation hypothesis, because the decrease in Number Flux from 4.5 km downwards is substantially larger than the decrease in Snow Flux over the same heights. The decrease of number (flux) relative to mass (flux) is exactly what is expected from aggregation. The overall decrease of Snow Flux with height could be explained by sublimation of the ice particles in subsaturated air (included in our calculations in

(5)), or through some process where large particles become significantly smaller (e.g. collisional breakup; not included in (5)). Nevertheless, these properties also support rapid aggregation in this part of the cloud.

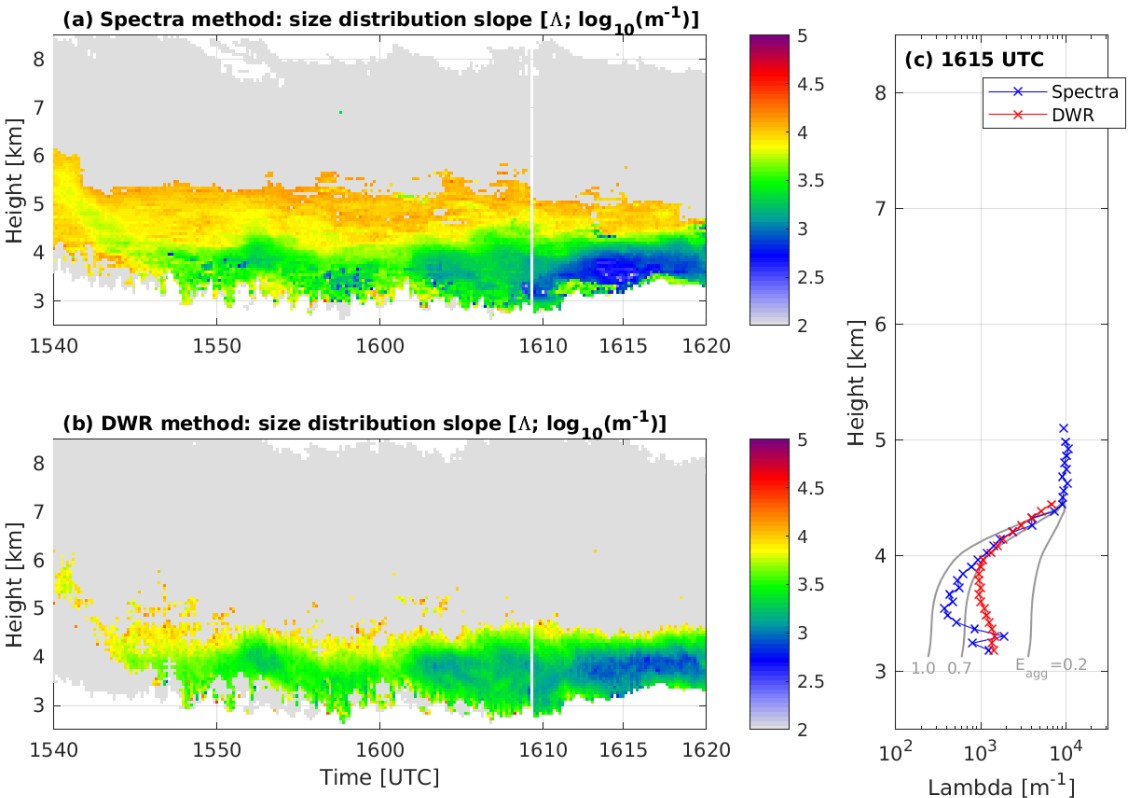

**Figure 6.** Time-height plots of $\Lambda$, the slope of the ice particle size distribution derived from the a) multi-wavelength Doppler spectra method and b) the dual-wavelength ratio (DWR) method. The grey regions mark areas of the cloud where no retrieval of $\Lambda$ was possible. See text for details. Panel c) shows a profile of values averaged over 2 minutes centred on 1615 UTC. The grey lines show the expected changes in $\Lambda$ for three different values of aggregation efficiency (1.0, 0.7, 0.2), assuming the ice particle size distribution at 4.5 km evolves due to aggregation alone. The $E_{agg} = 1.0$ CSF curve assumes a constant snow-flux from 4.5 km downwards.

## 6.1 Sensitivity to uncertainties in the retrieval

The retrieval of the properties of the ice particle size distribution is naturally sensitive to uncertainties in the input quantities. To determine to what extent our retrieval is sensitive to these uncertainties, the retrieval has been repeated with a range of different assumptions. The sensitivity analysis looks at three different aspects: 1) the impact of improperly aligning the Doppler spectra from the radars along the velocity axis; 2) the impact of improperly aligning the Doppler spectra based on reflectivity or

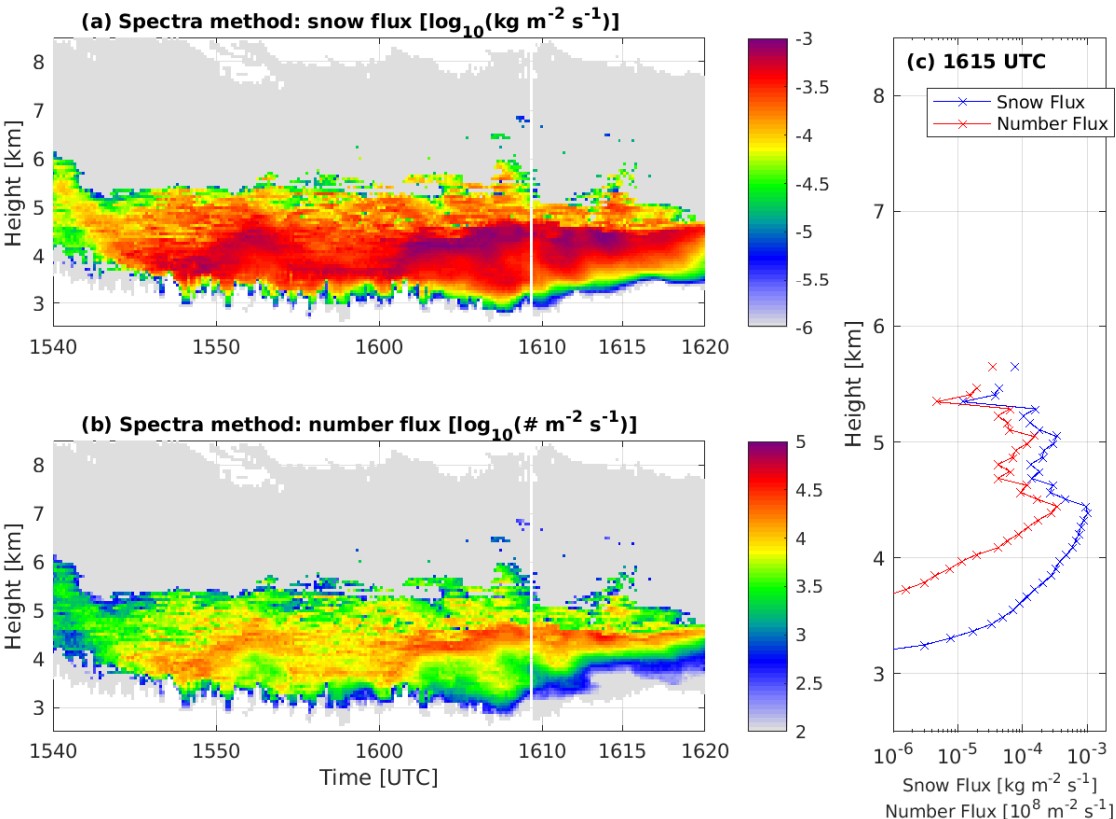

**Figure 7.** Time-height plots of the retrieved quantities of a) Snow flux and b) Number flux. These quantities are calculated for particles with retrieved diameter >0.75 mm only, and therefore underestimate the true snow and number flux. Panel c) shows the profile of these two quantities retrieved at 1615 UTC.

calibration errors; and 3) the impact of using a different mass-diameter relationship in the retrieval. The details of the different sensitivity tests are given in Table 2.

Figure 8 shows how the retrieved ice particle size distribution at 4.15 km altitude and at 1615 UTC varies under the different uncertainty assumptions. There are some large variations in the maximum ice particle diameters retrieved - in particular for

5   uncertainties related to changing the velocity (Aspect 1; blue lines) and also in the number concentration retrieved at a particular diameter can vary by an order of magnitude. However, the overall character of the size distribution is usually unchanged, and when the characteristic slope of the size distribution $\Lambda$ is calculated, it is largely insensitive to the uncertainties.

This insensitivity of $\Lambda$ to these uncertainties can be seen in Fig. 9. In panels a-d of this figure, the vertical profile of $\Lambda$ at 1615 UTC is shown for each of the different uncertainties. This can be compared with Fig. 6c and the retrieved $\Lambda$ profile from

10   the unperturbed setup is plotted in black on panels a-d. Although there is some variation of $\Lambda$ for the different uncertainty assumptions, the vertical profile continues to show rapid decreases of $\Lambda$ with height down from 4.5 km, consistent with large

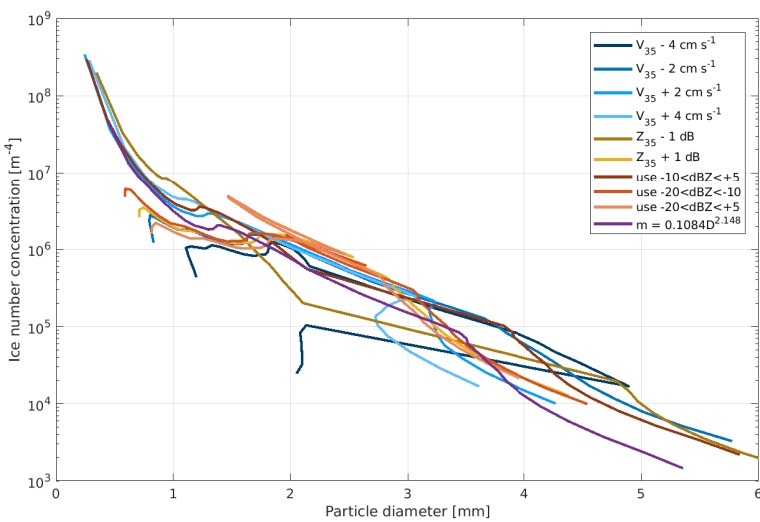

**Figure 8.** The ice particle size distribution at 4.15 km (equivalent to Figure 5i) under various uncertainty assumptions. See Table 2 for details about the uncertainties included in these retrievals.

**Table 2.** Details of the changes to the retrieval input or parameters for the sensitivity testing

| Aspect | Name | Description |
|---|---|---|
| 1 | $V_{35} - 4$ cm s$^{-1}$ | Doppler spectra from 35 GHz radar shifted to the left by two velocity bins (0.04 m s$^{-1}$) |
| 1 | $V_{35} - 2$ cm s$^{-1}$ | Doppler spectra from 35 GHz radar shifted to the left by one velocity bin (0.02 m s$^{-1}$) |
| 1 | $V_{35} + 2$ cm s$^{-1}$ | Doppler spectra from 35 GHz radar shifted to the right by one velocity bin (0.02 m s$^{-1}$) |
| 1 | $V_{35} + 4$ cm s$^{-1}$ | Doppler spectra from 35 GHz radar shifted to the right by two velocity bins (0.04 m s$^{-1}$) |
| 2 | $Z_{35} - 1$ dB | 1 dB subtracted from $Z_{35}$ and $sZ_{35}$ |
| 2 | $Z_{35} + 1$ dB | 1 dB added to $Z_{35}$ and $sZ_{35}$ |
| 2 | use $-10 < \text{dBZ} < +5$ | Calibration of $Z_{35}$ and $Z_{94}$ to match $Z_3$ in regions where $-10 < Z_3 < +5$ dBZ |
| 2 | use $-20 < \text{dBZ} < -10$ | Calibration of $Z_{35}$ and $Z_{94}$ to match $Z_3$ in regions where $-20 < Z_3 < -10$ dBZ |
| 2 | use $-20 < \text{dBZ} < +5$ | Calibration of $Z_{35}$ and $Z_{94}$ to match $Z_3$ in regions where $-20 < Z_3 < +5$ dBZ |
| 3 | $m = 0.1048D^{2.148}$ | Replaces the mass-diameter relationship from Brown and Francis (1995) with that from Heymsfield (2013) for aggregate snowflakes |

aggregation efficiency values (Fig. 9e). The largest deviation is seen for the uncertainty where $Z_{35}$ and $sZ_{35}$ are reduced by 1 dB. This change results in larger $\Lambda$ values at all heights due to a reduction of DWR$_{35/94}$ by 1 dB. The lower sDWR results in the retrieved particle diameters being smaller such that the largest particles have lower number concentrations and therefore a steeper slope is diagnosed. Nevertheless, the change of $\Lambda$ with height for this uncertainty is also consistent with rapid aggregation. Therefore we conclude that none of the uncertainties assessed substantially change the conclusion that

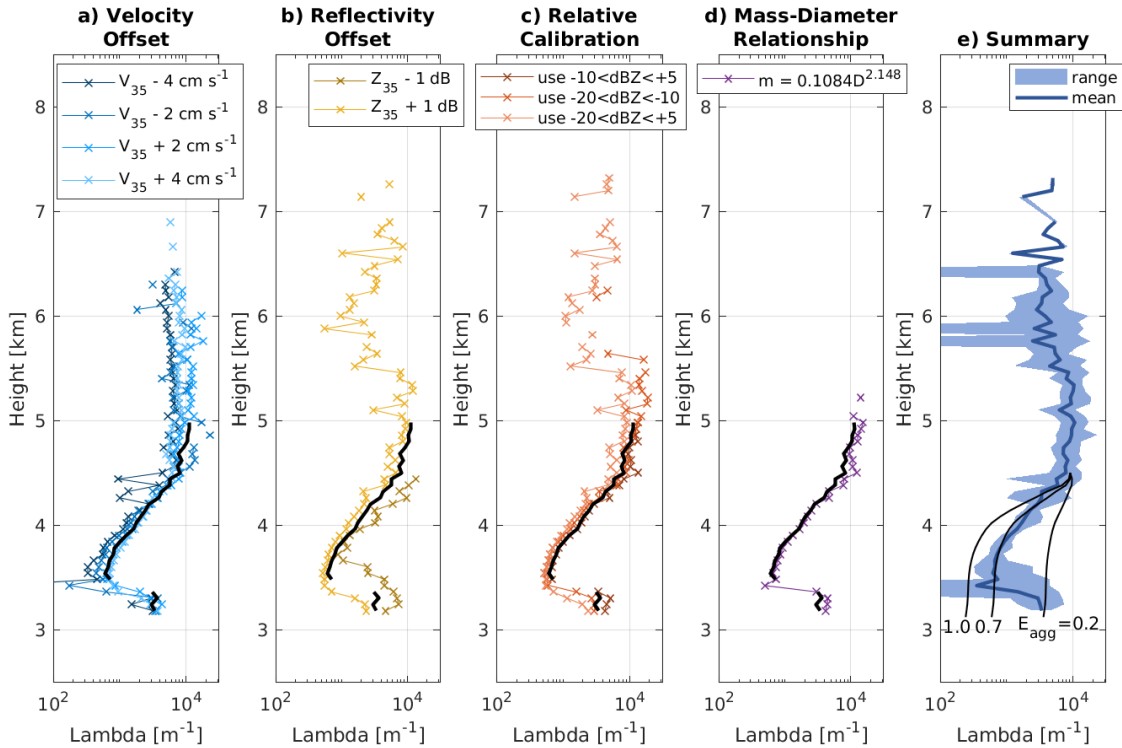

**Figure 9.** Panels a-d show the vertical profile of $\Lambda$ at 1615 UTC (equivalent to Figure 6c) under various uncertainty assumptions. Panel e) shows the mean (solid line) and range (shaded region) as a function of height for all individual retrievals shown in panels a-d. The mean is calculated from the base-10 logarithm of the plotted values. The unperturbed retrieval is plotted on panels a-d in black for comparison. The theoretical curves for changes of $\Lambda$ with height due to aggregation and starting from 4.5 km altitude for $E_{agg}$ values of 1.0, 0.7, 0.2 are shown on panel e, as in Fig. 6c.

aggregation is likely the dominant mechanism for changing the ice particle size distribution from 4.5 km downwards between 1545 and 1620 UTC.

The estimation of the aggregation efficiency value is largely dependent on the mass-size and velocity-size relationships used, because these control the values $a$, $b$, $c$ and $d$ which are the main terms in (5) determining the change of $\Lambda$ with height. $b$ and $d$ also contribute substantially to change of $I_1$. These values are, however, relatively well constrained. First, (5) is totally insensitive to $a$, because it appears only once and is cancelled out because it also contributes to the mass flux $\chi_f$. Second, $b = 1.9$ is known for this case (Stein et al., 2015). Therefore no sensitivity exists to the choice of mass-size relationship. $c$ and $d$ have been estimated from the power-law fits as part of the retrieval process, and are quite well constrained within the aggregation region (Fig. 10).

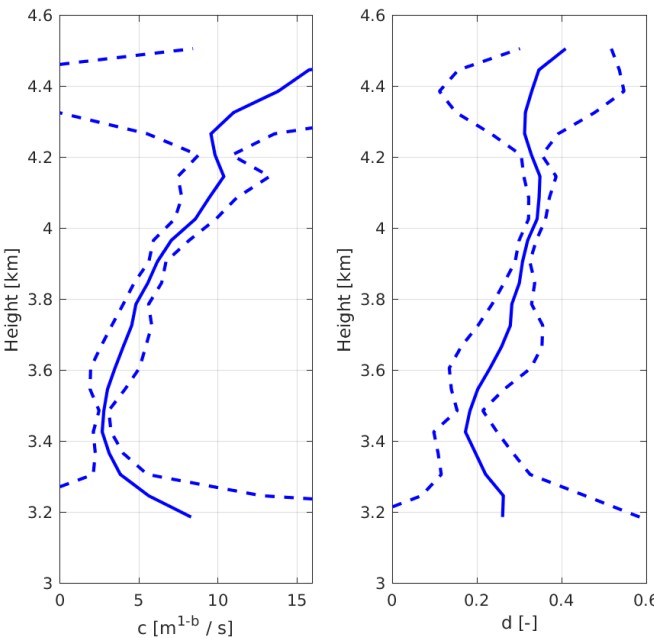

**Figure 10.** Vertical profiles of $c$ and $d$ from the power-law fits to the velocity and diameter retrievals between 1614 and 1616 UTC. The mean (solid) and the spread approximated by the standard deviation of c- and d-values in time at each height (dashed) are shown.

## 7 Conclusions

We have shown that the use of radar Doppler spectra data from three co-located, vertically-pointing radars at frequencies of 3, 35 and 94 GHz can produce estimates of the ice particle size distribution and be used to identify and explore processes such as aggregation. Different radar reflectivity for different radar frequencies shows evidence that there are particles present that are large enough that they are no longer within the Rayleigh scattering regime. Using the Doppler spectra from the three radars, we can determine the size and estimate the number of these ice particles.

In the case presented in this paper, we identify a region where the 35 to 94 GHz dual-wavelength ratio (DWR) increases rapidly with decreasing height, indicative that large ice particles are forming quickly. We have ruled out vapor deposition as the cause of these large particles, because that process is too slow. Similarly the rapid growth is not a result of riming because there was no evidence of significant liquid water. We therefore argue that these large particles, up to 5 mm in diameter, are a result of aggregation. Our observations are consistent with theoretical calculations of ice particle size distribution evolution resulting from purely aggregation. In this case an aggregation efficiency around 0.7 fits the observations.

Aggregation as the cause of the rapid growth of ice particles is supported by the consistent and narrow range of heights over which this change occurs. The rapid aggregation occurs just below the region where the Doppler spectra is bi-modal, indicating the presence of small, newly-formed ice particles. It appears that the small ice particles forming at approximately 5.3 km

(−15.4 °C), and appearing clearly in the Doppler spectra at 4.8 km (Fig. 5d), grow into dendritic ice particles at temperatures around −15 °C and either aggregate with other similarly formed particles, or initiate aggregation with pre-existing ice particles falling through this layer. The aggregation initiated by these particles is then evident by the large particles present at 4.1 km, which could not have been formed by vapor deposition or riming.

These observations of rapid aggregation at temperatures around −15 °C add support to cloud chamber studies (Connolly et al., 2012; Hobbs et al., 1974), which also suggest that aggregation at around −15 °C is much more efficient that at other temperatures. The resulting changes to the ice particle size distribution through this aggregation process strongly affect many microphysical process rates (e.g. vapour deposition, sedimentation velocity, further aggregation) and therefore failure to capture these aggregation regions in models can lead to significantly errors in the simulated cloud fields.

This multi-wavelength Doppler spectra technique shows the ability to determine the size distribution of ice particles in large portions of ice clouds simultaneously. Previously, ice particle sizes have been determined using ice particle sizing instruments attached to aircraft, which suffer from two issues: small sample sizes and shattering of large ice particles on the instrument inlet, resulting in many small particles in the sample volume and leading to unreliable estimates of both large and small ice particle concentrations (Westbrook and Illingworth, 2009; Korolev et al., 2011). Therefore further studies of cloud microphysical
structure and processes using this method are encouraged.

For the benefit of future studies, we give some advice here for achieving the best results. To achieve reliable, quantitative results from this method, the radars need to be very precisely pointed vertically. We find that mis-pointing by 0.2° is sufficient to resolve a non-negligible contribution from the horizontal wind in the Doppler spectra, which adds extra challenges to comparing the spectra from the three radars. Ideally the three radars should also have the same beamwidth; spectral broadening
increases for wider beams and again makes comparing spectra from different radars more challenging, especially in the tails of the spectra where the largest DWR values are expected. Despite these challenges, we have shown that this technique enables the generation of ice particle size distributions from remote sensing data. We were unable to make reliable retrievals in regions of strong turbulence (e.g. due to instability created by sublimation) because the assumption that the spectra was unchanged over the 10-second averaging window was violated. Although no low-level clouds were present on this day, the technique
to cross-calibrate the radars near cloud top enables the retrieval to be performed even when (supercooled-)liquid clouds or rain are partially attenuating the radar signal at lower levels. Retrievals of this type have the potential to benefit the cloud microphysics community through both statistical sampling of clouds and aiding studies of individual processes, such as the aggregation process detailed in this paper. Further studies comparing the retrieved size distributions against data obtained from aircraft are currently being performed. One weakness of our current experimental setup is that we can only size particles larger
than 0.75 mm. Particles smaller than 0.75 mm are in the Rayleigh scattering regime at all three wavelength and therefore their size cannot be determined. The addition of an extra shorter wavelength (e.g. at frequencies of 150 or 220 GHz, as advocated by Battaglia et al. (2014)), would enable sizing of particles down to approximately 0.45 or 0.3 mm (for 150 and 220 GHz respectively). Such observations would provide a unique opportunity for increasing our understanding of cloud microphysics, both statistically and through process studies as demonstrated in this paper.

**Author contributions**

AB performed most of the data analysis, analyzed the radar data, and wrote the main manuscript text. CW conceived and led the research project, contributed code for the Westbrook scattering model and assisted with the radar data analysis. JN created the pre-processing code for the radar data. TS provided code and experience from previous work with the data. All authors
5  discussed the scientific findings and contributed to the final manuscript.

*Acknowledgements.* This research was funded by the Natural Environment Research Council, grant NE/K012444/1. The radar data used in this paper can be accessed at the Centre for Environmental Data Archival (www.ceda.ac.uk) or by contacting the authors. We are grateful to the staff at the Chilbolton Facility for Atmospheric and Radio Research for operating and maintaining the radars and, in particular, Alan Doo, Allister Mallett, Chris Walden, John Bradford, and Darcy Ladd for their assistance in collecting the triple-wavelength measurements.

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
