# Peer review of "Rapid ice aggregation process revealed through triple-wavelength Doppler spectra radar analysis"

_Atmospheric Chemistry and Physics, 2018_

## Referee Comment (RC1) · Anonymous Referee #1 · 2 Oct 2018

General Comments

The paper focuses on the identification of a rapid aggregation layer within an ice cloud. In order to do so, an innovative algorithm for the retrieval of snow particle size distribution (PSD) is developed. The algorithm leverages on the synergies of multi-frequency and Doppler observations from vertically pointing radar systems. The retrieved evolution of snowflake sizes is connected to microphysical processes through a modeling approach. It is concluded that neither depositional growth nor riming can explain alone the rapid increase in snow size and aggregation must play a major role, moreover, the expected sticking efficiency must be larger than what was previously published in

dedicated laboratory experiments.

The paper puts emphasis on the properties of the rapid aggregation layer and in particular to the value of the aggregation efficiency (Eagg). This would entitle the paper to be published on ACP given the importance of this process in ice clouds. However I am not sure if the reported conclusions are sufficiently supported by scientific evidence. In particular, I am concerned about the number of strict assumptions that have been made throughout the text, the lack of independent validation of the results and the very short duration of the single event selected to support the conclusions about Eagg value. By contrast the paper propose a very interesting and innovative way to use vertically pointing radar to retrieve the properties of particle size distributions. As best of my knowledge, this is the first retrieval of the size-resolved PSD using radars, which would allow to drop the assumptions about PSD shape that are necessary in bulk approaches. The presented methodology deserves a much more detailed description than the one presented in the text and a profound analysis of the sensitivity of the method to the various assumptions that have been made. After such revisions, I would definitely recommend to publish it, but I would suggest to consider a different journal such as AMT given the shift of the focus of the paper.

Given all of my concerns, I recommend to consider the paper to be published after a major revision.

Specific comments

1) Given the centrality of the concept for the entire manuscript I would suggest to give a definition of Eagg in the introduction section. This also to prevent potential confusion, given by the non-unique nomenclature used in this field where different efficiencies might mean different things (e.g. collection efficiency, sticking efficiency). Finally, this would help understanding the reasoning behind the last paragraph of section 6 and Figure 5, where Eagg is inferred from the vertical gradient of the slope parameter of the PSD.

2) I am not convinced by the statement about Connolly et al. (2012) at lines 31-34 of page 2. By looking at Fig. 14 in the original paper I would agree on the fact that Eagg is between 0.4 and 0.9 at -15 C because that is the confidence interval reported in the plot. For the very same reason I would say that it is between 0 and 0.5 for the other temperatures. Claiming that it is always below 0.2 might be an exaggerated statement.

3) In section 2.1 the non-microphysical sources of differential reflectivity (DWR) are accounted for. These source of retrieval error are compensated by making the radar reflectivity to match in the Rayleigh-scattering part of the cloud and applying the same adjustment to the whole profile (lines 4-5 of page 4). This method is strictly valid only for radar miscalibration and attenuation by radome or wet antenna; for height dependent sources of differential attenuation (e.g. atmospheric gas absorption, liquid in the cloud) this method causes an overcompensation of the higher frequency reflectivities at lower level (in particular W-band). Attenuation due to atmospheric gases depends on the density and humidity profile of the atmosphere and can adds up to 2.5 dB at the top of the cloud at midlatitude; under this condition the overcompensation caused by the radar cross-calibration at 3-4 km could be easily in the order of 1 dB. I suggest to compensate for atmospheric gases absorption profile before making the radar-cross calibration, or, at least, estimate the W-band absorption profile for the analyzed case by using either a weather model or a radiosonde profile and report it either in the paper or in supplementary material.

4) 35- and 94- GHz radars are declared to be off-zenith by 0.2 and 0.15 deg in opposite directions (lines 15-16 of page 4). This causes a contamination of the doppler signal from horizontal wind component which is then corrected by shifting the spectra by constant values (line 2 page 5). The authors acknowledge the fact that this procedure is imperfect and consider the matching of the resulting spectra to be good in Figure 4. However it is not described how these numbers (mispointing angle and Doppler velocity correction) are obtained. I personally can hardly see how a composite relative shift of just 0.1 m/s would have affected the matching in Figure 4. The comparison in Figure 4

would have taken benefit from the inclusion of the 3 GHz spectrum which is considered to be well aligned. Also the 'goodness' of the matching is both affected by velocity and power shifts: a small residual differential attenuation would have caused the spectra to look non-aligned; given the fact that there might be still a differential attenuation issue here (see my point number 3) the matching is potentially flawed by this residual error. I suggest to include the source of the mispointing angles and Doppler shifts numbers.

5) The PSD and v-D retrieval method is very roughly explained in pages 6 and 7. After several readings I understood that it assumed a unique relation between sDWR and the snowflake size. This relation is likely to be very specific of the assumed scattering model and mass-size relation. A plot showing this relation for a certain number of scattering models and m-D functions will help the reader understanding the methodology applied and gives an indication of the expected uncertainty due to the related assumptions.

6) Moreover, for the scattering model it is assumed Westbrook (2006, 2008a) since it has been found to closely match observation in the multifrequency space [Stein 2015]. However, the scattering model from Leinonen and Szyrmer (2015) has also been found to match the observation (unpublished on a peer review journal, but included in conference proceedings http://www.isac.cnr.it/~ipwg/meetings/bologna-2016/Bologna2016_Orals/3-8_Westbrook.pdf) It would be very interesting to see the results from this different scattering model. Being a detailed DDA model one does not have to assume the m-D relation but can simply take particles masses and sizes from the database, achieving a better consistency of the results.

7) The particles that are sampled within each Doppler bin are likely to have different sizes. Is the model considering only one particle size per Doppler bin? This is potentially a significant source of error when large particles are present. Large particles are expected to fall roughly at the same velocity for many different sizes, thus dv/dD « 1, by contrast the backscattering signal given by those particles is very different. Assuming that in fast-falling doppler bins (i.e. v>1m/s) there are snow particles of just one size is

not a valid assumption even at for doppler systems with a very high spectral resolution.

8) Considering the number of correction, a sensitivity analysis of the algorithm with respect to the input data is essential. Assuming 1 dB uncertainty in radar reflectivity and 1 or 2 velocity bins uncertainty in the doppler spectra will already give a good indication of the robustness of the algorithm. It will be particularly interesting to see how this translates into uncertainties in the retrieved PSDs (panels c, f and i of figure 4) and the profile of fitted scale parameter lambda in figure 5.

9) The result of 'rapid aggregation' is obtained by comparing the relative potential of various snow growth processes, concluding that only aggregation is capable to give such rapid change in PSDs scale parameter lambda. I think that the potential given by the PSDs retrieval is here underutilized. Given the full PSD and the m-D and v-D relations one can calculate interesting bulk quantities such as particle number concentration (Nt) and ice water content (IWC) and their vertical fluxes (particle flux Nf and snow rate SR). It will be extremely interesting to see time-height plots of this quantities in connection with the results in figure 4 and 5. For instance, positive variation of Nf and SR should be seen in connection with the newly developed mode in fig4d. This analysis would also help in the identification of the significant aggregation process. Depositional growth and riming are in fact expected to increase SR leaving Nf unchanged (unless newly nucleating particles are present). Aggregation has the distinctive effect of decreasing Nf leaving SR unchanged and this should appear in the suggested plots.

10) At line 27 of page 12 it is mentioned that the methodology described in Mitchell (1988) has been used to model the evolution of lambda parameter, however it is not specified the exact model used. It is surprising that the formula for the depositional growth rate from Pruppacher and Klett (1978) has been fully reported and not this. This is potentially a serious issue regarding the reproducibility of the results. Additionally, I think that a better explanation of the model used will give the reader a better understanding of the other variables that influence the PSD evolution due to aggregation such as particle sizes, velocity differences and total number concentration.

11) The conclusions about the value of the Eagg are supported by only a 2 minute average profile during one event. I would, at least, model the evolution of the lambda parameter for other times during the same event, or, even better, model more events.

Technical corrections

12) When presenting the state of the art of Doppler/multi-frequency radar retrievals at line 20-25 of page 2, I suggest to consider some recent studies like Chase et al. (2018) or Leinonen et al. (2018) in the discussion.

13) The choice of the colormap used in figures 2, 3 and 5 is particularly unfortunate. There is an apparent overlap of light-blue colors for different values that makes the interpretation of the figures more difficult than it should be. In figure 4b there are vast areas of the cloud where I cannot say if the DWR is either +1 or -1 dB. In figure 5 the mapping from the colorplot to the profile is made even more difficult by the fact that the profile as been cut from the panel with a white line; here I also suggest to indicate the profile with a thin rectangle around instead of the white line.

14) Figure 4 – Personally I would swap the axis in panels b, e, h. This would put velocity on x-axis, matching the concept on panels a, d and g. Also it appears that DWR is rather a function of velocity and not the opposite (see in particular panel e). That is a personal preference and I would leave to the authors the decision.

References

Chase, R. J., Finlon, J. A., Borque, P., McFarquhar, G. M., Nesbitt, S. W., Tanelli, S., et al. (2018). Evaluation of triple‐frequency radar retrieval of snowfall properties using coincident airborne in situ observations during OLYMPEX. Geophysical Research Letters, 45, 5752–5760. https://doi.org/10.1029/2018GL077997

Connolly, P. J., Emersic, C., and Field, P. R.: A laboratory investigation into the aggregation efficiency of small ice crystals, Atmos. Chem. Phys., 12, 2055-2076, https://doi.org/10.5194/acp-12-2055-2012, 2012.

Leinonen, J., and W. Szyrmer (2015), Radar signatures of snowflake riming: A modeling study, Earth and Space Science, 2, 346–358, doi:10.1002/2015EA000102

Leinonen, J., Lebsock, M. D., Tanelli, S., Sy, O. O., Dolan, B., Chase, R. J., Finlon, J. A., von Lerber, A., and Moisseev, D.: Retrieval of snowflake microphysical properties from multi-frequency radar observations, Atmos. Meas. Tech. Discuss., https://doi.org/10.5194/amt-2018-73, in review, 2018.

Mitchell, D.L., 1988: Evolution of Snow-Size Spectra in Cyclonic Storms. Part I: Snow Growth by Vapor Deposition and Aggregation. J. Atmos. Sci., 45, 3431–3451, https://doi.org/10.1175/1520-0469(1988)045<3431:EOSSSI>2.0.CO;2

Stein, T. H. M., C. D. Westbrook, and J. C. Nicol (2015), Fractal geometry of aggregate snowflakes revealed by triple‐wavelength radar measurements, Geophys. Res. Lett., 42, 176–183, doi:10.1002/2014GL062170

---

## Referee Comment (RC2) · Anonymous Referee #2 · 15 Oct 2018

**1   Summary**

This manuscript proposes a new algorithm to retrieve the particle size distribution (PSD) from vertically pointing Doppler profilers at 3 frequencies, using the spectral dual-wavelength ratio and not the ratio of integrated reflectivity values as in previous work. This algorithm is then applied in the context of the study of a given cloud, to investigate the dominant microphysical processes taking place and explaining the measured Doppler spectra. Rapid aggregation appears to be the best candidate among various processes to explain the observed behavior.

[Figure]

**2 Recommendation**

The algorithm and the application for microphysical interpretation that are presented in this manuscript are innovative and relevant. A "direct" PSD estimation without any assumption about its mathematical functional form is promising and useful. But there are also a number of assumptions that are required to run this "inversion", and they are not all clearly described and discussed. It is hence difficult to understand in which framework this approach can be safely used, and the example presented in this manuscript remains rather specific. The manuscript is pleasant to read with quality illustrations. Overall, I am convinced that this manuscript presents innovative and original material that are worth publication, but after having addressed the issues listed below.

**3 General comments**

1. Information about the methodological side is missing: no detailed/exhaustive description of the proposed PSD spectral retrieval algorithm is provided, making it difficult to check or reproduce for instance. I suggest the authors to add detailed description (including equations and so on) of the different steps of the algorithm.

2. The case study is too limited (40 min of a single cloud) to derive general insights beyond the demonstration that the proposed method works, at least for one cloud. I understand the difficulty to expand the analysis, but this example is too limited in itself (see below).

3. From a more general point of view, I have the feeling that this manuscript "oscillates" between the two Copernicus journals AMT and ACP, between a more methodological point of view (e.g. the retrieval algorithm) and a more meteorological point of view (case study of rapid aggregation in a cloud). So in the end, the reader is somehow frustrated: on the one hand, the paper proposes

a new retrieval method (AMT side), but does not provide enough description of this method for the reader to implement it; on the other hand the case study is too limited to gain any general insights into cloud microphysics (ACP side). I am fine with the authors choosing ACP, but I would strongly recommend to add more explanations about the proposed retrieval technique, as well as more discussion about the limitations and the conditions in which this approach is valid. There is some content in this direction in the conclusion (p.15, l.7-11) but only the verticality and the beam width are discussed, not the requirements in terms of turbulence, (supercooled-)liquid water or not, the geographical representativity, etc.

**4  Specific comments**

1. P.8, l.2: optimal with respect to what? Which fitting method is employed to estimate the power-law parameters?

2. P.8, l.2: why using a power law between vertical terminal velocity and the size?

3. P.8, l.10: so the 3 GHz spectra are used "only" for large particles? If so, the proposed approach is essentially dual-frequency. Should the title be adapted?

4. P.10, l.26-27: what are the plausible mechanisms to explain the generation of these new ice particles? Maybe it was mentioned somewhere but if so, I missed it.

5. P.11, l.25-27: is a SNR threshold applied prior to run the retrieval, in order to filter out the noisy values?

---

## Referee Comment (RC3) · Anonymous Referee #3 · 16 Oct 2018

The authors present a method to quantify the aggregation process and retrieve the ice particle size distribution using three co-located radars. They showed that aggregation causes a rapid (less than 10 minutes) growth of ice particles from 0.75 mm to 5 mm in maximum size. They speculate that the dendrites dominate at -15 C with large aggregation efficiency (approximated to be near unity). Although the results are important and the manuscript is interesting, there are multiple issues that have to be addressed before the manuscript can be accepted for publication. My suggestions are explained below.

General comments:

[Figure]

- How do you distinguish between the ice particles and water drops? In pg 5, ln 21, you said that your case is an ice cloud. Elsewhere you mentioned that there was no water drop in the cloud. However, a mixed-phase cloud is probable in this temperature range. Fig. 3 shows that the temperature in the presence of cloud ranges from 0 to -40 C. Between -38 and 0 C, super-cooled water drops co-exist with ice particles (Rosenfeld and Woodley, 2000), and there is a great chance of water contamination. It is important to address this, and explain how you detect water drops and exclude them. Alternatively, is it possible to quantify the ratio of liquid water content to ice water content?

- There is no comparison between your retrieval and direct measurements of size spectra, because there was no in-situ measurement available for your case. It is true that disagreements exist between various in-situ probes (see also Fig. 6 in Cotton et al., 2010), but still it is not certain if your retrieved size spectra would be more accurate. It would be good to cite any study that compared retrieved size spectra with direct observations. In any case, such caveat (no comparison between your retrieval and in-situ measurements) should be explained in the manuscript, and should be mentioned in the abstract and conclusions.

- The Brown and Francis (1995) mass-size relation has an important issue: it's not realistic for size smaller than 100 microns, since it gives ice particle mass larger than that of a sphere. See Erfani and Mitchell (2016) and their Fig. 1. I understand that you do not detect particle smaller than 0.75 mm, but it is important to address this issue for the readers who might use Brown and Francis mass-size relation. In addition, the readers will become aware of the more recent mass-size relations.

- Your radar is unable to detect particles smaller than 0.75 mm. This means that your retrieved data is not able to approximate the vast majority of particle number density or dN/dD (because small particles dominate the number concentration or N; again see Fig. 6 in Cotton et al., 2010). How does that affect your calculations? Since the calculation of number concentration is an important part of your paper, you should

highlight this limitation (no detection for size less than 0.75 mm) and its consequences in the abstract and conclusions.

Specific comments:

- abstract, ln 5: Did you calculate the mean size change by aggregation?

- abstract, ln 11: Any evidence to support this? I understand that this is suggested based on previous studies. If yes, it should be mentioned explicitly: "Based on previous study, we suggest ..."

- pg 2, ln 7: By "cloud microphysical properties", do you mean individual ice particle properties such particle size or mass?

- pg 2, ln 14: Please add at least one example (with a reference) on how different size spectra affect the relative importance of microphysical processes.

- pg 2, ln 15: Please add a reference.

- pg 2, last paragraph: It is good to cite Keith and Saunders (1989), since they performed experiments and measured the aggregation efficiency for various shapes and sizes. They showed that the aggregation efficiency ranges between 0.3 and 0.85 for planar snow crystals depending on the particle size.

- pg 3, Section 2: Please add proper references for each radar and for the near-field correction method. Overall, this section doesn't have sufficient citations and I can see only 2 references in the whole section.

- pg 4, paragraph starting at ln 14: Have you tried to correct the direction of 2 radars and make a few measurements, and then compare with the previous measurements?

- Table 1: Right now, it is not mentioned anywhere in the manuscript.

- pg 5, ln 26: Are these temperatures measured by radiosonde?

- pg 5, ln 28, Change to "Figure 2b".

- pg 5, ln 30: Was the Westbrook model initialized for the same cloud?

- Fig. 2b an 2c: The explanation of Fig. 2b in the manuscript is not enough. What is the physical interpretation of such difference between the two radars. Also, the explanation of Fig. 2c is missing in the manuscript.

- Fig.3 and 4: You explained Fig. 4 in the manuscript earlier than Fig. 3, so please switch these figures.

- pg 6, ln 6: Briefly define the scattering model. Also, do you mean individual ice particle or a bulk property such as mean size or median size?

- pg 7, ln 2: See the general comment regarding mass-size relation. It would be good to cite Erfani and Mitchell (2016) since they explained recent mass-size relations.

- pg 7, ln 1-4: The steps 2 and 3 need to include the relationships you used to relate various variables.

- Fig. 4: When the x-axis says "particle diameter", do you mean the maximum size of each particle, or did you calculate the sphere-equivalent diameter? Moreover, the explanation of panels b-e-h is missing in the manuscript.

- pg 10, ln 15-16: When particle sizes grow, but their fall speed does not increases, this is a sign of branching and aggregation rather than riming. See Locatelli and Hobbs (1974).

- pg 10, ln 17-24: Combine all these lines into one paragraph.

- pg 10, ln 32: Doppler spectra is not bi-modal in Fig. 4a. Do you mean Fig. 4b?

- pg 10, ln 33: Why are such small particles a result of nucleation and not growth by vapor deposition, or a secondary ice production (such as fragmentation of ice particles)? Elsewhere you assumed the small particles in the bi-modal spectra are the result of vapor deposition. Any evidence on the mechanism responsible for the increase in small particles?

- pg 11, ln 1: You say the aggregation causes ice particles to grow larger and fall faster, but aggregate fall speed does not grow by size. See Locatelli and Hobbs (1974) and their Fig. 12. They also provide fall speed-size relations for various ice particle shapes (including dendrites and aggregates).It's good to cite this paper, and also it would be great if you fit their relation to your data and calculate the R-squared.

- pg 11, ln 3-4: This can be a sign of aggregation.

- pg 11, ln 11: This is an exponential function. Moreover, I assume D and dN/dD are known in this equation. How did you calculate N0? It is important to explain this in the paper. It seems that the value of slope is dependent on the calculation of N0. Furthermore, do you use such distribution to relate size to radar reflectivity? Your size spectra do not include small particles. Since the number of small particles contributes significantly to the number concentration, how did this affect your calculations?

- pg 11, ln 13: This is a qualitative comparison. Have you looked at the difference between Fig. 5a and 5b? From Fig. 5c, it seems the agreement between the 2 slopes is not excellent. Note that this is a logarithmic axis and I can see the red line can be larger by a factor of 1.5.

- pg 12, ln 10-12: How did you calculate F (ventilation coefficient) and K (thermal conductivity)?

- pg 12, ln 26-27: The vapor deposition and riming do not change the total number of ice particles (N), but they do change the number of ice particles within each size bin (dN/dD). Please clarify that the rate of change in size is not the same for all sizes. As an example for riming: riming collision efficiency is a strong function of ice particle size, so larger ice particles would grow faster due to riming. See Wang and Ji (2000) and their Fig. 7; might be good to cite this paper.

- pg 12, ln 27-28: Please refer to the proper equation number in Mitchell (1988).

- pg 14, last paragraph: This statement is suited for the Introduction and can be moved

near the end of Introduction as the motivation for your study.

- pg 15, first paragraph: See my general comments on the lack of comparison with in-situ measurements; I agree the issues exists in directly measuring the particle size and concentration, but still it is unclear how your method has better accuracy. In addition, please cite Cotton et al. (2010) when explaining the disagreements in the in-situ measurements of ice particles.

References:

Cotton, R., Osborne, S., Ulanowski, Z., Hirst, E., Kaye, P. H., & Greenaway, R. S. (2010). The ability of the Small Ice Detector (SID-2) to characterize cloud particle and aerosol morphologies obtained during flights of the FAAM BAe-146 research aircraft. Journal of Atmospheric and Oceanic Technology, 27(2), 290-303, https://doi.org/10.1175/2009JTECHA1282.1.

Erfani, E. and Mitchell, D. L. (2016). Developing and bounding ice particle mass- and area-dimension expressions for use in atmospheric models and remote sensing, Atmos. Chem. Phys., 16, 4379-4400, https://doi.org/10.5194/acp-16-4379-2016.

Keith, W. D. and Saunders, C. P. R. (1989). The collection efficiency of a cylindrical target for ice crystals, Atmos. Res., 23, 83-95, https://doi.org/10.1016/0169-8095(89)90059-8.

Locatelli, J. D., & Hobbs, P. V. (1974). Fall speeds and masses of solid precipitation particles. Journal of Geophysical Research, 79(15), 2185-2197, https://doi.org/10.1029/JC079i015p02185.

Rosenfeld, D. and Woodley, W. L. (2000). Deep convective clouds with sustained supercooled liquid water down to -37.5 C, Nature, 405, 440–442, doi:10.1038/35013030.

Wang, P. K., & Ji, W. (2000). Collision efficiencies of ice crystals at low–intermediate Reynolds numbers colliding with supercooled cloud droplets: A numerical study. Journal of the atmospheric sciences, 57(8), 1001-1009, https://doi.org/10.1175/1520-

0469(2000)057%3C1001:CEOICA%3E2.0.CO;2.

---

## Referee Comment (RC4) · P Connolly (Referee) · 1 Nov 2018

**Rapid ice aggregation process revealed through triple-wavelength Doppler spectra radar analysis**

Paul Connolly, University of Manchester, November 1, 2018

**General**

This is a very well put together study, combining data from three co-located, vertically pointing radars to quantify aggregation efficiencies in the atmosphere.

It is the first attempt to retrieve the ice particle size distribution from multi-frequency Doppler spectra observations. It is satisfying to see that these results, in the main, corroborate our chamber observations.

The presentation is very good and there are no major issues. I recommend publication, but would like to see more information on the fall speed relations used and perhaps an assessment of how the results depend on mass-dimension relations.

As I have worked on similar problems before I wanted to see whether I could reproduce the findings from the information available in the paper, to check that my interpretation is consistent with the main findings in the paper. My reasons for doing this are to demonstrate how others may interpret your data, and to check that my interpretation is correct.

I present this alternate analysis, in a separate section below.

**Specific Comments**

Page 3, line 25 – sentence begins with "because".

Page 7: Brown and Francis to convert between mass and size. The Brown and Francis (1995) relation is for ice crystals in cirrus clouds. There are more up to date mass-size relations that are published so I was curious if you have tried these, and whether a different assumption affects the results

Page 8: velocity power law – you don't give examples of the fit parameters here, which makes it more difficult for others to understand your data. I wonder if you could give some example figures, or statistics of the fit parameters (the a and b coefficients).

Figure 4: convincing plot. Just a comment: I am surprised that the spectral reflectivity of the middle plot and bottom plot extends to just above 1.5 m/s , whereas the size distributions are much broader for the lower layer. Is this because the larger particles in the lower layer are less dense so that their fall speed saturates with increasing particle size?

**Alternate Analysis**

Validation: in order to better understand figure 4 I thought I would do a consistency check. I digitised your data from plots in figure 4 c, f, and i

First I wanted to calculate the mass flux, to see if this was roughly in-line with that expected and to see whether it was approximately conserved between levels. As you are aware the mass flux should be conserved in diffusional growth is not important.

I used the Brown and Francis (1995) relation to convert particle diameter to mass (as you have done)

$$m = 0.0185D^{1.9}$$

And, as you have not given the coefficients for the velocity-size relation, I have used a fall speed relation from Wang and Chang (1993):

$$v = 6.96D^{0.33}$$

My analysis is shown in Figure 1. I have calculated the mass flux at the top, middle, and bottom of the cloud presented in your figure 4. The values I have calculated are as follows

|  | top | middle | bottom |
|---|---|---|---|
| Mass flux $10^{-4}$ (kg m-2 s-1) | 4.12 | 2.39 | 4.81 |

[Figure]

**Figure 1. Analysis of your figure 4. Data points are taken from your figure 4, lines are exponential fits. Text shows the calculated mass flux. Colours are as follows: red (top of cloud); green (middle); blue (bottom of cloud).**

We should expect that the mass flux increases if the particles grow by vapour diffusion, or decreases if the particles evaporate. If vapour diffusion is not important we should observe that the mass flux is conserved. Here we see approximately a factor of 1.7 reduction in the mass flux in the middle of the cloud. I suspect that these numbers are within the expected retrieval errors (or errors in mass-dimension / velocity – dimension relations, but it would be useful if you could comment on this.

The fact that I have used a velocity – dimension power law that is not based on your observation may also be responsible for this too: another reason why it would be helpful to see your velocity-size relations.

Next I thought I would try the analysis of Mitchell (1988) to attempt to calculate the aggregation efficiency. The relevant equation is equation 16 in Mitchell (1988).

$$\frac{d\lambda}{dz} = -\frac{\pi E_a I_1 \chi_f \lambda^{\beta+b-1}}{4\beta a\alpha\Gamma(\beta+b+1)\Gamma(2\beta+b+1)}$$

which can be rearranged for $E_a$, the aggregation efficiency. Here, $\beta$=1.9, b=0.33, a=6.96, $\alpha$=0.0185 (SI units); $\lambda$ is the slope of the size-distribution; $\chi_f$ is the mass flux (the mass falling through an area per second); $\Gamma$ is the gamma function and $I_1$ is a definite integral to be calculated (see Ferrier et al 1994)

From the data in Figure 1 I was able to estimate $\frac{d\lambda}{dz}$ to be 7.9 (SI units); $\lambda$=6.42e3; and $\chi_f$=2.39e-4 (SI units) are based on values in the middle of the cloud.

I calculated the integral, $I_1$, as 37.89 – code can be provided on request – feel free to contact me.

From these numbers, and rearranging Mitchell's equation above, one can estimate Ea to be equal to approximately 0.4.

This number is not too far from what you have used, but it would be useful to understand where the differences arise – I think your estimate is a little higher. For instance on page 12 you say you also use Mitchell (1988); hence, I wondered whether you could go through the calculation in more detail. I suspect this is due to the power laws used for velocity – size, but it may also be due to errors in fitting slope and intercept parameters to the data for instance.

I was not sure whether you had taken into account diffusional growth either. Taking into account diffusional growth with increase the slope, so the aggregation efficiency will need to be higher than I have calculated to lead to the observed reduction in the slope.

Additionally my estimate of Ea=0.4 assumes the mass flux in the middle of the cloud to be 2.4e-4, which is low compared to the top and bottom. If I use the

higher mass flux 4.8e-4 (the value I calculated from your data at cloud base), in the calculation, the corresponding Ea is approximately 0.2.

In addition I thought I would try and reproduce a plot similar to your figure 5c. My Figure 2 shows these simulations using aggregation efficiencies of 1, 0.4 and 0.2. The finding here is that lower values of the aggregation efficiency yield lambda values closer to your observations at the 4 km level. Again I think the reason for this discrepancy may be because my calculations have used a terminal fall speed power law that does not match the observations for small crystals. Since the calculations appear to be quite sensitive to the terminal fall speed relation it would be really useful if you could present the measured fall speed (and regression coefficients) you have used.

[Figure]

**Figure 2. Model simulation using the initial conditions taken from the top of the cloud in figure 4, using different values of the aggregation coefficient.**

Final word – I strongly support the statement about sizing particles down to 0.3mm, which would allow you to probe earlier stages of aggregation.

**References**

Brown, P R A, and P N Francis. 1995. "Improved Measurements of the Ice Water Content in Cirrus Using a Total-Water Probe" 12: 410–14.

Ferrier, B S. 1994. "A Double-Moment Multiple-Phase Four-Class Bulk Ice Scheme. Part I: Description" 51: 249–80.

Mitchell, D. 1988. "Evolution of Snow-Size Spectra in Cyclonic Storms. Part I: Snow Growth by Vapour Deposition and Aggregation" 45: 3431–51.

Wang, Chien, and Julius S. Chang. 1993. "A Three-Dimensional Numerical Model of Cloud Dynamics, Microphysics, and Chemistry: 1. Concepts and Formulation." *Journal of Geophysical Research* 98 (D8): 14827. doi:10.1029/92JD01393.

---

## Author Comment (AC1) · 15 Mar 2019

**We thank each of the reviewers for their helpful comments and criticism. We have modified the paper taking into account all of these points and feel that the revised paper is significantly improved and that our results are now presented more clearly and the findings are more robust. Here is a brief overview of the key changes made to the paper.**

- **Rewritten paper to be more focussed on the meteorologically interesting aspects of this case.**
- **Improved the clarity of the description of the new retrieval technique**
- **Inclusion of new equations defining the aggregation efficiency and the expected change to the lambda (slope of particle size distribution) with height due to aggregation from Mitchell (1988)**
- **Added more information about the assumptions made, their validity and a comprehensive sensitivity test to these assumptions (new section 6.1)**
- **New figures showing:**
  - **the relationship between diameter and DWR for the Westbrook (2006,2008) scattering model, and for particles from Leinonen and Moisseev (2015) (figure 1)**
  - **Snow flux and number flux as derived from our retrieval (figure 7)**
  - **Sensitivity of the particle size distribution to assumptions in the retrieval (figure 8)**
  - **Sensitivity of the change of lambda (slope of particle size distribution) to assumptions in the retrieval (figure 9)**
  - **Information about the statistical properties of the velocity-diameter power law fits made in the retrieval (figure 10)**

**We believe that the improved clarity and additional information now provided in the paper make our results more convincing.**

**Specific replies to the individual reviewers comments are below.**

Reviewer 1

General Comments

The paper focuses on the identification of a rapid aggregation layer within an ice cloud. In order to do so, an innovative algorithm for the retrieval of snow particle size distribution (PSD) is developed. The algorithm leverages on the synergies of multi-frequency and Doppler observations from vertically pointing radar systems. The retrieved evolution of snowflake sizes is connected to microphysical processes through a modeling approach. It is concluded that neither depositional growth nor riming can explain alone the rapid increase in snow size and aggregation must play a major role, moreover, the expected sticking efficiency must be larger than what was previously published in dedicated laboratory experiments.

The paper puts emphasis on the properties of the rapid aggregation layer and in particular to the value of the aggregation efficiency (Eagg). This would entitle the paper to be published on ACP given the importance of this process in ice clouds. However I am not sure if the reported conclusions are sufficiently supported by scientific evidence. In particular, I am concerned about the number of strict assumptions that have been made throughout the text, the lack of independent validation of the results and the very short duration of the single event selected to support the conclusions about Eagg value. By contrast the paper propose a very interesting and innovative way to use vertically pointing

radar to retrieve the properties of particle size distributions. As best of my knowledge, this is the first retrieval of the size-resolved PSD using radars, which would allow to drop the assumptions about PSD shape that are necessary in bulk approaches. The presented methodology deserves a much more detailed description than the one presented in the text and a profound analysis of the sensitivity of the method to the various assumptions that have been made. After such revisions, I would definitely recommend to publish it, but I would suggest to consider a different journal such as AMT given the shift of the focus of the paper.

Given all of my concerns, I recommend to consider the paper to be published after a major revision.

**We thank the reviewer for their comments, and have ensures that in the revised manuscript there is a clearer description of the retrieval method and a sensitivity analysis to determine the impact of uncertainties in the retrieval. Although the exact details of the particle size distribution showed variability in the sensitivity testing, the conclusion that the broadening of the size distribution is a result of aggregation appears robust. We have therefore decided to keep the paper focussed on the Meteorological aspects of this event and keep the paper within ACP-scope.**

Specific comments

1) Given the centrality of the concept for the entire manuscript I would suggest to give a definition of Eagg in the introduction section. This also to prevent potential confusion, given by the non-unique nomenclature used in this field where different efficiencies might mean different things (e.g. collection efficiency, sticking efficiency). Finally, this would help understanding the reasoning behind the last paragraph of section 6 and Figure 5, where Eagg is inferred from the vertical gradient of the slope parameter of the PSD.

**Thank you for the suggestion. We have added the formal definition of aggregation efficiency based on Mitchell (1988)'s aggregation kernel to the paper – see equation 1 and the supporting text.**

2) I am not convinced by the statement about Connolly et al. (2012) at lines 31-34 of page 2. By looking at Fig. 14 in the original paper I would agree on the fact that Eagg is between 0.4 and 0.9 at -15 C because that is the confidence interval reported in the plot. For the very same reason I would say that it is between 0 and 0.5 for the other temperatures. Claiming that it is always below 0.2 might be an exaggerated statement.

**We have revised the text to clarify that the "best estimate" of E_agg is below 0.2, except at -15C.**

3) In section 2.1 the non-microphysical sources of differential reflectivity (DWR) are accounted for. These sources of retrieval error are compensated by making the radar reflectivity to match in the Rayleigh-scattering part of the cloud and applying the same adjustment to the whole profile (lines 4-5 of page 4). This method is strictly valid only for radar miscalibration and attenuation by radome or wet antenna; for height dependent sources of differential attenuation (e.g. atmospheric gas absorption, liquid in the cloud) this method causes an overcompensation of the higher frequency reflectivities at lower level (in particular W-band). Attenuation due to atmospheric gases depends on the density and humidity profile of the atmosphere and can adds up to 2.5 dB at the top of the cloud at midlatitude; under this condition the overcompensation caused by the radar cross-calibration at 3-4 km could be easily in the order of 1 dB. I suggest to compensate for atmospheric gases absorption profile before making the radar-cross calibration, or, at least, estimate the W-band absorption profile for the analyzed case by using either a weather model or a radiosonde profile and report it either in the paper or in supplementary material.

**We understand the reviewers comment and have considered this issue. However, it should be noted that the overwhelming majority of attenuation from liquid water and gases (and therefore also differential attenuation) occurs in the lower troposphere, below the level of clouds we are analysing in this paper. Therefore, our application is analogous to that of radome attenuation (where the attenuation occurs before the first target of interest). To further support this argument, we have calculated the attenuation from water vapor as a function of height (using a nearby radiosonde profile), and determined that 85% of the attenuation occurs below cloud base. The remaining attenuation would contribute to about 0.15 dB difference of 35 and 94 GHz power. Such an offset in power translates to about 60 microns in difference of the retrieved particle size at both cloud base and cloud top. Therefore, the conclusions drawn about the rapid aggregation occurring in this cloud are not affected by the attenuation correction we have made. However, the reviewer is correct to point out that there will be some situations where this method will not work correctly and we have added a sentence to the paper stating this to discourage future researchers from blindly applying this method.**

**Additionally, the fact that the spectra at all 3 frequencies overlay one another nicely in the upper levels of the cloud suggest the relative calibration works well. If there were significant non-Rayleigh effects, they would be more prominent in the spectra which would show sZ94 too high for small (slow falling) particles, and sZ94 too low for large (fast falling particles). We don't see that behaviour. In order to address the reviewers concerns, we have added different possible definitions of the "Rayleigh-scattering regime" in which we match the reflectivity from the 3 radars as part of the sensitivity testing. Our findings appear robust to which range of dBZ values are chosen.**

4) 35- and 94- GHz radars are declared to be off-zenith by 0.2 and 0.15 deg in opposite directions (lines 15-16 of page 4). This causes a contamination of the doppler signal from horizontal wind component which is then corrected by shifting the spectra by constant values (line 2 page 5). The authors acknowledge the fact that this procedure is imperfect and consider the matching of the resulting spectra to be good in Figure 4. However it is not described how these numbers (mispointing angle and Doppler velocity correction) are obtained. I personally can hardly see how a composite relative shift of just 0.1 m/s would have affected the matching in Figure 4. The comparison in Figure 4 would have taken benefit from the inclusion of the 3 GHz spectrum which is considered to be well aligned. Also the 'goodness' of the matching is both affected by velocity and power shifts: a small residual differential attenuation would have caused the spectra to look non-aligned; given the fact that there might be still a differential attenuation issue here (see my point number 3) the matching is potentially flawed by this residual error.

I suggest to include the source of the mispointing angles and Doppler shifts numbers.

**Following the reviewers suggestion, we have also added the 3 GHz spectra to the plots in Figure 4 (now Figure 5). In accordance to our reply to the previous point, the goodness of fit of all three spectra (in regions where we expect them to be identical, e.g. figure 4a (now figure5a)) provided by making these corrections to the measured Doppler velocity suggest that it is necessary and beneficial. Contrary to the reviewers expectations, a mis-alignment of the spectra by even 0.02 m/s can show substantial differences in the retrieved particle size distribution. This difference is because the spectra is relatively "steep", sZ changing rapidly as a function of velocity. Hence, slightly misaligned spectra result in artificially enhanced sDWR values, either for small velocities, or for large velocities. A relative offset of 0.1 m/s renders the retrieval useless – see the retrieved size spectra when such an uncertainty is added in the new sensitivity analysis. The importance of the velocity offset for the retrieval has been clarified in the revised paper.**

**A description of how the offsets and pointing angle errors were calculated was not included in the original paper because the cause of the offsets is not of practical significance, but the fact that we have corrected the data to account for them is important. For completeness we have added a short description to the paper and also here:**

**The velocity differences as a function of height were determined be assessing the mean Doppler velocities from the three radars individually. The correlation of these offsets with the atmospheric wind profile (determined from ECMWF forecast fields) enabled an estimation of the pointing angle errors. We include these values in the paper only to highlight to future researchers that rather small pointing angle errors can significantly affect the retrievals and therefore care should be taken to ensure that the radars are pointed as accurately as possible.**

**The goodness of matching of the spectra is indeed affected by both velocity and power offsets. However, these are easy to separate and correct for independently. The velocity offset can be determined through correlation of the Doppler spectra while power offsets can be determined by integrating received power across the spectra. In fact, we used bulk methods to determine the offsets (mean velocity difference between radars and total reflectivity differences – as discussed above) and these also worked well to ensure the Doppler spectra are well matched.**

5) The PSD and v-D retrieval method is very roughly explained in pages 6 and 7. After several readings I understood that it assumed a unique relation between sDWR and the snowflake size. This relation is likely to be very specific of the assumed scattering model and mass-size relation. A plot showing this relation for a certain number of scattering models and m-D functions will help the reader understanding the methodology applied and gives an indication of the expected uncertainty due to the related assumptions.

**The reviewer is correct that this section was not sufficiently clear in the submitted version. The text has been improved to add clarity and the figure suggested has been added to the paper (new Figure 1) to highlight both the method and the characteristics of the scattering model used.**

6) Moreover, for the scattering model it is assumed Westbrook (2006, 2008a) since it has been found to closely match observation in the multifrequency space [Stein 2015]. However, the scattering model from Leinonen and Szyrmer (2015) has also been found to match the observation (unpublished on a peer review journal, but included in conference proceedings http://www.isac.cnr.it/ ~ ipwg/meetings/bologna2016/Bologna2016_Orals/3-8_Westbrook.pdf) It would be very interesting to see the results from this different scattering model. Being a detailed DDA model one does not have to assume the m-D relation but can simply take particles masses and sizes from the database, achieving a better consistency of the results.

**While comparison with other scattering models is indeed interesting, we present plenty of evidence in this paper, and also in Stein et al. 2015, that this scattering model is suitable for this case. The Leinonen and Szyrmer (2015) scattering model is for rimed aggregates, and we have already stated in the paper that there is no evidence for riming, either in terms of the presence of supercooled liquid water or the characteristic behaviour of the pair of DWR values presented in Stein et al (2015). However, the scattering calculations in the presentation that you link to are indeed for unrimed aggregates – these have very similar characteristics to the Westbrook (2006) scattering model used. We have added points to Figure 1 to highlight the similarities of the two scattering models.**

**Additionally, it should be noted that we do not require a mass-size relationship to relate sDWR and particle size – the retrieval method for size is independent of the mass-size relationship.**

7) The particles that are sampled within each Doppler bin are likely to have different sizes. Is the model considering only one particle size per Doppler bin? This is potentially a significant source of error when large particles are present. Large particles are expected to fall roughly at the same velocity for many different sizes, thus dv/dD « 1, by contrast the backscattering signal given by those particles is very different. Assuming that in fast-falling doppler bins (i.e. v>1m/s) there are snow particles of just one size is not a valid assumption even at for doppler systems with a very high spectral resolution.

**We agree with the reviewer that the relationship between particle size and particle velocity is likely not a one-to-one monotonic function. This assumption and the limitations have been further discussed in the revised paper compared to the original version. However, as a first attempt at using such a technique to retrieve the particle size distribution without fixed assumptions about the shape of the distribution - we need to make some assumption here. It may well be that a refined approach could yield more accurate or robust results; however, this would require a-priori information on the statistical variability in V for a given D, which is poorly constrained at present, and therefore we leave that for future work.**

8) Considering the number of correction, a sensitivity analysis of the algorithm with respect to the input data is essential. Assuming 1 dB uncertainty in radar reflectivity and 1 or 2 velocity bins uncertainty in the doppler spectra will already give a good indication of the robustness of the algorithm. It will be particularly interesting to see how this translates into uncertainties in the retrieved PSDs (panels c, f and i of figure 4) and the profile of fitted scale parameter lambda in figure 5.

**A sensitivity analysis incorporating uncertainties of +/- 1 dB to the Doppler spectra, a shift in velocity space of up to +/- 0.04 m/s, which range of reflectivity values are used for attenuation correction and an additional mass-diameter relationship has been added. The impact of the uncertainties of the size distribution (equivalent to figure 4i) and of the vertical profile of Lambda (equivalent to figure 5) have been added. The sensitivity analysis adds confidence to our argument that aggregation is the primary driver of change to the size distribution in the lower region of the cloud. Furthermore, it shows the (negative) impact on the particle size distribution retrieval when the Doppler spectra are not suitable matched – suggesting that our matching methodology is in fact reliable (albeit imperfect).**

9) The result of 'rapid aggregation' is obtained by comparing the relative potential of various snow growth processes, concluding that only aggregation is capable to give such rapid change in PSDs scale parameter lambda. I think that the potential given by the PSDs retrieval is here underutilized. Given the full PSD and the m-D and v-D relations one can calculate interesting bulk quantities such as particle number concentration (Nt) and ice water content (IWC) and their vertical fluxes (particle flux Nf and snow rate SR). It will be extremely interesting to see time-height plots of this quantities in connection with the results in figure 4 and 5. For instance, positive variation of Nf and SR should be seen in connection with the newly developed mode in fig4d. This analysis would also help in the identification of the significant aggregation process. Depositional growth and riming are in fact expected to increase SR leaving Nf unchanged (unless newly nucleating particles are present). Aggregation has the distinctive effect of decreasing Nf leaving SR unchanged and this should appear in the suggested plots.

**We thank the reviewer for this suggestion, which has resulted in the addition of a new figure showing the number flux and snow flux throughout the interesting part of the observed cloud. We believe that the coherence in these plots adds support that our retrieval is stable and as the revierer suggested – the vertical profile of the number and snow flux do add support to the rapid**

**aggregation hypothesis. Although the snow flux decreases with height (presumably due to sublimation), the number flux decreases with height significantly faster.**

10) At line 27 of page 12 it is mentioned that the methodology described in Mitchell (1988) has been used to model the evolution of lambda parameter, however it is not specified the exact model used. It is surprising that the formula for the depositional growth rate from Pruppacher and Klett (1978) has been fully reported and not this. This is potentially a serious issue regarding the reproducibility of the results. Additionally, I think that a better explanation of the model used will give the reader a better understanding of the other variables that influence the PSD evolution due to aggregation such as particle sizes, velocity differences and total number concentration.

**We apologise for this oversight and have now added the full equation from Mitchell (1988) to our paper. Furthermore, because of the additional analysis of the snow flux added, we decided that the constant snow-flux with height assumption was not valid, and instead used the retrieved snow flux profile in the calculation. We have maintained the constant snow-flux profile for E_agg of 1.0 in the paper for comparison.**

11) The conclusions about the value of the Eagg are supported by only a 2 minute average profile during one event. I would, at least, model the evolution of the lambda parameter for other times during the same event, or, even better, model more events.

**While we understand the reviewers request for the analysis of more data, we focus in this paper on the most microphysically interesting part of the cloud. We do not claim to make any general quantification of the aggregation efficiency, but rather to say that in this instance the observations are consistent with a large aggregation efficiency and the new retrieval has helped identify this process.**

**We have collected radar data from several days and believe that we have seen similar events within that dataset. We have focussed on this case study as it is microphysically interesting enough to serve as an example of the new retrieval technique and the insights into cloud microphysical processes that it can provide. The analysis of further events, where similar features were observed, are underway. The analysis of these events is not ready to be included in this paper and will hopefully be published separately (although the research grant for this work has now expired).**

Technical corrections

12) When presenting the state of the art of Doppler/multi-frequency radar retrievals at line 20-25 of page 2, I suggest to consider some recent studies like Chase et al. (2018) or Leinonen et al. (2018) in the discussion.

**Thanks. These more recent papers have been added to this discussion.**

13) The choice of the colormap used in figures 2, 3 and 5 is particularly unfortunate. There is an apparent overlap of light-blue colors for different values that makes the interpretation of the figures more difficult than it should be. In figure 4b there are vast areas of the cloud where I cannot say if the DWR is either +1 or -1 dB. In figure 5 the mapping from the colorplot to the profile is made even more difficult by the fact that the profile as been cut from the panel with a white line; here I also suggest to indicate the profile with a thin rectangle around instead of the white line.

**Actually the white line in figure 5 is because of missing data. The profile is at 1615 UTC, which has been emphasised by adding the time to the title of panel c and as an additional label on the x-axis of panels a and b.**

**We have experimented with different colour scales; however, none are able to cover the full data range while enabling values to be read from the figure accurately. It is because of this that we added vertical profiles of the values at 1615 UTC to figures 3, 6 and 7.**

14) Figure 4 – Personally I would swap the axis in panels b, e, h. This would put velocity on x-axis, matching the concept on panels a, d and g. Also it appears that DWR is rather a function of velocity and not the opposite (see in particular panel e). That is a personal preference and I would leave to the authors the decision.

**This is a good suggestion and has been changed in the new version of the figure.**

Reviewer2

1 Summary

This manuscript proposes a new algorithm to retrieve the particle size distribution (PSD) from vertically pointing Doppler profilers at 3 frequencies, using the spectral dual-wavelength ratio and not the ratio of integrated reflectivity values as in previous work. This algorithm is then applied in the context of the study of a given cloud, to investigate the dominant microphysical processes taking place and explaining the measured Doppler spectra. Rapid aggregation appears to be the best candidate among various processes to explain the observed behavior.

2 Recommendation

The algorithm and the application for microphysical interpretation that are presented in this manuscript are innovative and relevant. A "direct" PSD estimation without any assumption about its mathematical functional form is promising and useful. But there are also a number of assumptions that are required to run this "inversion", and they are not all clearly described and discussed. It is hence difficult to understand in which framework this approach can be safely used, and the example presented in this manuscript remains rather specific. The manuscript is pleasant to read with quality illustrations. Overall, I am convinced that this manuscript presents innovative and original material that are worth publication, but after having addressed the issues listed below.

**We thank the reviewer for their supportive comments. We have developed the manuscript in response to the reviewers comments (as detailed below) with a more rigorous analysis of the uncertainties and a better description of the assumptions and retrieval technique. As a result, we feel this has made the paper much more convincing and the results robust.**

3 General comments

1. Information about the methodological side is missing: no detailed/exhaustive description of the proposed PSD spectral retrieval algorithm is provided, making it difficult to check or reproduce for instance. I suggest the authors to add detailed description (including equations and so on) of the different steps of the algorithm.

**The algorithm is actually relatively simple, but we acknowledge that the description could have made the process clearer to the reader. We have added additional text and clarifications and believe that the revised manuscript provides a better basis for reproducing this method.**

2. The case study is too limited (40 min of a single cloud) to derive general insights beyond the demonstration that the proposed method works, at least for one cloud. I understand the difficulty to expand the analysis, but this example is too limited in itself (see below).

**The focus on this case is because of the interesting, abrupt in height but consistent in time, appearance of large DWR values lower in the cloud. We do not try to claim that this is a common feature, or that the aggregation efficiencies derived are common to other clouds/cloud types. As discussed in the paper, data on the aggregation efficiency is rare and there is a large spread in the reported values. Therefore we believe the additional insight from this paper to be valuable to the community. We present an analysis of what is occurring at this time and height in this cloud, and show how this method can be useful to investigate processes in such clouds. Further study of other clouds using the same method is underway at a preliminary stage, and we hope that this study and retrieval technique will provide a foundation to analyse the variability of aggregation**

**efficiency in clouds in a systematic way and to evaluate how it depends on temperature, relative humidity etc.**

3. From a more general point of view, I have the feeling that this manuscript "oscillates" between the two Copernicus journals AMT and ACP, between a more methodological point of view (e.g. the retrieval algorithm) and a more meteorological point of view (case study of rapid aggregation in a cloud). So in the end, the reader is somehow frustrated: on the one hand, the paper proposes a new retrieval method (AMT side), but does not provide enough description of this method for the reader to implement it; on the other hand the case study is too limited to gain any general insights into cloud microphysics (ACP side). I am fine with the authors choosing ACP, but I would strongly recommend to add more explanations about the proposed retrieval technique, as well as more discussion about the limitations and the conditions in which this approach is valid.

There is some content in this direction in the conclusion (p.15, l.7-11) but only the verticality and the beam width are discussed, not the requirements in terms of turbulence, (supercooled-)liquid water or not, the geographical representativity, etc.

**Based on this comment and that from other reviewers, we have tried to focus more on the meteorological aspects within the paper, but at the same time clarifying adding some more details about the retrieval technique.**

4 Specific comments

1. P.8, l.2: optimal with respect to what? Which fitting method is employed to estimate the power-law parameters?

**The word optimal has been removed. The power-law was estimated by fitting a linear best-fit line to the logarithm of the values.**

2. P.8, l.2: why using a power law between vertical terminal velocity and the size?

**The power law has been used because 1) it is easily differentiable and 2) it is common in microphysical scaling relationships.**

3. P.8, l.10: so the 3 GHz spectra are used "only" for large particles? If so, the proposed approach is essentially dual-frequency. Should the title be adapted?

**The 3-GHz is essential for the attenuation correction of the radars (because it is used to identify the Rayleigh-scattering part of the cloud and provide a reference). It is used in the sizing of particles larger than 2.2 mm (which can be done for both 35 and 94 GHz, and should provide the same answer). It is furthermore useful to help identify the correct scattering model to use – as done in Stein et al (2015). Therefore, although some aspects only employ dual-frequency techniques, the complete retrieval is dependent on all three radars. The 3GHz spectra has additionally been added to Figure 5 to enable comparison between all three radars.**

4. P.10, l.26-27: what are the plausible mechanisms to explain the generation of these new ice particles? Maybe it was mentioned somewhere but if so, I missed it.

**We have not speculated on the generation mechanism because we have no data that will help determine or rule out any mechanism.**

5. P.11, l.25-27: is a SNR threshold applied prior to run the retrieval, in order to filter out the noisy values?

**Yes, noisy data points are filtered out and details have been added to the text.**

Reviewer 3

The authors present a method to quantify the aggregation process and retrieve the ice particle size distribution using three co-located radars. They showed that aggregation causes a rapid (less than 10 minutes) growth of ice particles from 0.75 mm to 5 mm in maximum size. They speculate that the dendrites dominate at -15 C with large aggregation efficiency (approximated to be near unity). Although the results are important and the manuscript is interesting, there are multiple issues that have to be addressed before the manuscript can be accepted for publication. My suggestions are explained below.

General comments:

- How do you distinguish between the ice particles and water drops? In pg 5, ln 21, you said that your case is an ice cloud. Elsewhere you mentioned that there was no water drop in the cloud. However, a mixed-phase cloud is probable in this temperature range. Fig. 3 shows that the temperature in the presence of cloud ranges from 0 to -40 C. Between -38 and 0 C, super-cooled water drops co-exist with ice particles (Rosenfeld and Woodley, 2000), and there is a great chance of water contamination.

It is important to address this, and explain how you detect water drops and exclude them. Alternatively, is it possible to quantify the ratio of liquid water content to ice water content?

**Mixed-phase clouds are possible in this temperature range; however, we are confident that the liquid water content in this cloud is negligible. Firstly, the microwave radiometer instrument does not detect any significant liquid water. Second, analysis of the radar Doppler spectra does not show any evidence of low reflectivity drops at small fall velocities (although they could be too small to be detectable). Third, the evidence of the pairs of DWR-values shown in Stein et al. (2015) are consistent with aggregates, and inconsistent with rimed particles – which suggests that there isn't a lot of supercooled water present. These clarifications are all included in the revised paper.**

- There is no comparison between your retrieval and direct measurements of size spectra, because there was no in-situ measurement available for your case. It is true that disagreements exist between various in-situ probes (see also Fig. 6 in Cotton et al., 2010), but still it is not certain if your retrieved size spectra would be more accurate. It would be good to cite any study that compared retrieved size spectra with direct observations. In any case, such caveat (no comparison between your retrieval and in-situ measurements) should be explained in the manuscript, and should be mentioned in the abstract and conclusions.

**We do not try to argue that our method is necessarily more accurate than in-situ measurements. However, the advantage is that we can make continuous measurements and multiple heights simultaneously and see the evolution of the size distribution. It is unfortunate that there is no in-situ data available, and such a comparison is part of the planned future work.**

- The Brown and Francis (1995) mass-size relation has an important issue: it's not realistic for size smaller than 100 microns, since it gives ice particle mass larger than that of a sphere. See Erfani and Mitchell (2016) and their Fig. 1. I understand that you do not detect particle smaller than 0.75 mm, but it is important to address this issue for the readers who might use Brown and Francis mass-size relation. In addition, the readers will become aware of the more recent mass-size relations.

**We agree that the Brown and Francis (1995) mass-diameter relationship is not physical for small sizes. However, such a failing does not affect our retrieval because it is not possible to reliably size**

**any particles smaller than 0.75 mm, and certainly not down to the 100 micron scale. We also note that Brown and Francis do address this issue in their original paper and for these reasons it seems unnecessary to include a repetition of that information in our paper.**

**As part of the newly added sensitivity analysis, uncertainty to the mass-diameter relationship is estimated. We also have evidence from the Stein et al. (2015) paper that the exponent (1.9) used in the Brown and Francis mass-diameter is consistent with the observations on this day.**

- Your radar is unable to detect particles smaller than 0.75 mm. This means that your retrieved data is not able to approximate the vast majority of particle number density or dN/dD (because small particles dominate the number concentration or N; again see Fig. 6 in Cotton et al., 2010). How does that affect your calculations? Since the calculation of number concentration is an important part of your paper, you should highlight this limitation (no detection for size less than 0.75 mm) and its consequences in the abstract and conclusions.

**Perhaps it was not clearly written in the paper, but it is incorrect to say that the radars cannot detect particles smaller than 0.75 mm diameter, but rather that their size cannot be determined reliably. We have attempted to clarify any possible misunderstanding in the revised paper, including a statement in the abstract that the size distribution can only be estimated for particles >0.75mm in diameter. The radars detect all particles of all sizes (assuming that there is enough total signal to differentiate that from background noise). While you are correct that the retrieval of the number concentration is important for the size distribution – we only attempt to retrieve the size distribution where sufficiently large particles exist. We could extrapolate back to small sizes from the fitted size distribution to determine an approximate number of small particles, but we have no need to do this for our study.**

Specific comments:

- abstract, ln 5: Did you calculate the mean size change by aggregation?

**Note that we are not referring to the mean size here (an advantage of our spectral technique). We argue that almost all of the size change is due to aggregation. We explain in the next sentence in the paper that the increase in size is shown to be consistent with aggregation when Eagg=0.7.**

- abstract, ln 11: Any evidence to support this? I understand that this is suggested based on previous studies. If yes, it should be mentioned explicitly: "Based on previous study, we suggest …"

**There is no direct evidence of this; however, such a process would be consistent with large aggregation efficiencies.**

- pg 2, ln 7: By "cloud microphysical properties", do you mean individual ice particle properties such particle size or mass?

**Yes, but not only individual ice particles properties, also bulk properties such as ice water content or equivalent properties for liquid water.**

- pg 2, ln 14: Please add at least one example (with a reference) on how different size spectra affect the relative importance of microphysical processes.

**We have added examples of why vapor deposition, riming and aggregation are affected by the particle size distribution. Details have been added to the text. Furthermore we have added numerous references to the importance of particle size distributions on correctly simulating different cloud types (page 3, line 19-25).**

- pg 2, ln 15: Please add a reference.

**The sentence reads "Another important application is to provide observations with which numerical models can be evaluated and their parameterizations improved." We do not think any reference is necessary here.**

- pg 2, last paragraph: It is good to cite Keith and Saunders (1989), since they performed experiments and measured the aggregation efficiency for various shapes and sizes. They showed that the aggregation efficiency ranges between 0.3 and 0.85 for planar snow crystals depending on the particle size.

**Thanks, we were previously unaware of this paper and we have now added this reference to the discussion**

- pg 3, Section 2: Please add proper references for each radar and for the near-field correction method. Overall, this section doesn't have sufficient citations and I can see only 2 references in the whole section.

**We have added the references for the three radars. The near-field correction was derived empirically by comparison of the 3 and 35 GHz radar profiles in low-reflectivity (Rayleigh scattering) ice clouds and is discussed briefly in Stein et al. (2015). This section is describing the data that we collected, and therefore additional citations would not be relevant here.**

- pg 4, paragraph starting at ln 14: Have you tried to correct the direction of 2 radars and make a few measurements, and then compare with the previous measurements?

**The errors in the pointing angles were only identified through the analysis in this paper. We could therefore not correct the pointing angles in time to observe this cloud.**

- Table 1: Right now, it is not mentioned anywhere in the manuscript.

**This has now been corrected.**

- pg 5, ln 26: Are these temperatures measured by radiosonde?

**The temperatures were from a nearby radiosonde and also from ECMWF forecasts.**

- pg 5, ln 28, Change to "Figure 2b".

**The reference to "Figure 2" is correct since we refer to both the time series of reflectivity and DWR, which are shown in separate panels.**

- pg 5, ln 30: Was the Westbrook model initialized for the same cloud?

**The scattering model predicts the observed radar reflectivity based on characteristics of the ice aggregates generated by an idealised model of the aggregation process. It is therefore a statistical relationship and does not use any measurements of the cloud on this day. However, as shown by Stein et al. (2015), there is good agreement between the expected behaviour of the DWR pairs and that predicted by the Westbrook scattering model.**

- Fig. 2b an 2c: The explanation of Fig. 2b in the manuscript is not enough. What is the physical interpretation of such difference between the two radars. Also, the explanation of Fig. 2c is missing in the manuscript.

**Further explanation has been added of panels a and b, explaining the quantities further and adding brief physical interpretation. Panel c is now explicitly referenced within this discussion.**

- Fig.3 and 4: You explained Fig. 4 in the manuscript earlier than Fig. 3, so please switch these figures.

**The discussion of figure 3 has been moved earlier**

- pg 6, ln 6: Briefly define the scattering model. Also, do you mean individual ice particle or a bulk property such as mean size or median size?

**The discussion has been complimented by the new Figure 1 showing how the DWR values change as a function of particle diameter for the scattering model. The diameter is that of an individual particle.**

- pg 7, ln 2: See the general comment regarding mass-size relation. It would be good to cite Erfani and Mitchell (2016) since they explained recent mass-size relations.

**The general comment was noted. As a result of reviewers comments, the sensitivity to the choice of mass-diameter relationship is now explicitly included within the sensitivity testing. Erfani and Mitchell (2016) provide more accurate but more complex mass-diameter relationships, whereas the estimation of the aggregation efficiency from Mitchell (1988) requires a mass-diameter relationship of the form $m=aD^b$.**

- pg 7, ln 1-4: The steps 2 and 3 need to include the relationships you used to relate various variables.

**We have clarified this section to address these and other concerns.**

- Fig. 4: When the x-axis says "particle diameter", do you mean the maximum size of each particle, or did you calculate the sphere-equivalent diameter? Moreover, the explanation of panels b-e-h is missing in the manuscript.

**The particle diameter is the maximum dimension, following Westbrook et al. (2006) Discussion of panels b-e-h added.**

- pg 10, ln 15-16: When particle sizes grow, but their fall speed does not increases, this is a sign of branching and aggregation rather than riming. See Locatelli and Hobbs (1974).

**At this point in the manuscript we do not make any assertions about the processes involved.**

- pg 10, ln 17-24: Combine all these lines into one paragraph.

**Done**

- pg 10, ln 32: Doppler spectra is not bi-modal in Fig. 4a. Do you mean Fig. 4b?

**Yes, corrected. Thanks.**

- pg 10, ln 33: Why are such small particles a result of nucleation and not growth by vapor deposition, or a secondary ice production (such as fragmentation of ice particles)? Elsewhere you assumed the small particles in the bi-modal spectra are the result of vapor deposition. Any evidence on the mechanism responsible for the increase in small particles?

**You are correct, we do not know how these particles have formed, possibly through nucleation, shattering or other processes. Nucleated has been replaced by formed in the text.**

- pg 11, ln 1: You say the aggregation causes ice particles to grow larger and fall faster, but aggregate fall speed does not grow by size. See Locatelli and Hobbs (1974) and their Fig. 12. They also provide fall speed-size relations for various ice particle shapes (including dendrites and aggregates). It's good to cite this paper, and also it would be great if you fit their relation to your data and calculate the R-squared.

**This is not what the sentence states. It says that the sDWR increases for the larger particles (which are also the fastest falling particles), but not that they necessarily fall faster.**

- pg 11, ln 3-4: This can be a sign of aggregation.

**This is the argument that we make later in the paper. Here we only present the evidence.**

- pg 11, ln 11: This is an exponential function. Moreover, I assume D and dN/dD are known in this equation. How did you calculate N0? It is important to explain this in the paper. It seems that the value of slope is dependent on the calculation of N0. Furthermore, do you use such distribution to relate size to radar reflectivity? Your size spectra do not include small particles. Since the number of small particles contributes significantly to the number concentration, how did this affect your calculations?

**Both N0 and Lambda are determined by fitting a straight line to the size distribution data (D and log10 (dN/dD)).**
**Such a fixed size distribution shape is not part of the retrieval, which is one advantage of our method.**

- pg 11, ln 13: This is a qualitative comparison. Have you looked at the difference between Fig. 5a and 5b? From Fig. 5c, it seems the agreement between the 2 slopes is not excellent. Note that this is a logarithmic axis and I can see the red line can be larger by a factor of 1.5.

**No, we have not performed a quantitative comparison. However, there good agreement in the values and patterns shown by the two methods. The disagreement at lower heights possibly highlights a weakness in the DWR method, where a fixed size distribution shape is required. We have no expectation that these two (nearly-independent) methods would give such a good agreement on the trend of lambda with height. The absolute values are not particularly interesting for our purposes.**

- pg 12, ln 10-12: How did you calculate F (ventilation coefficient) and K (thermal conductivity)?

**Suitable values for the appropriate temperature range were chosen.**
**(K=0.024, Rogers & Yau, chapter 7;**
**F=0.65 + 0.44*0.6^0.33 Re^0.5;**
**Re = rho * D * V(D) / dynamic_viscosity;**
**dynamic_viscosity is in the range 1.512E-5 … 1.862E-5, depending on temperature.**
**A clearer description has been added to the text.**

- pg 12, ln 26-27: The vapor deposition and riming do not change the total number of ice particles (N), but they do change the number of ice particles within each size bin (dN/dD). Please clarify that the rate of change in size is not the same for all sizes. As an example for riming: riming collision efficiency is a strong function of ice particle size, so larger ice particles would grow faster due to riming. See Wang and Ji (2000) and their Fig. 7; might be good to cite this paper.

**We have chosen to remove this statement from the paper as it is not key to the argument that aggregation is the main process acting in this cloud and adds unnecessary confusion. We present**

**sufficient evidence in the rest of the paper that aggregation is important and that deposition and riming are not the key processes involved.**

- pg 12, ln 27-28: Please refer to the proper equation number in Mitchell (1988).

**The full equation is now reproduced in the paper.**

- pg 14, last paragraph: This statement is suited for the Introduction and can be moved near the end of Introduction as the motivation for your study.

**Agreed. This paragraph of text has been moved**

- pg 15, first paragraph: See my general comments on the lack of comparison with in-situ measurements; I agree the issues exists in directly measuring the particle size and concentration, but still it is unclear how your method has better accuracy. In addition, please cite Cotton et al. (2010) when explaining the disagreements in the in-situ measurements of ice particles.

**Again, see the above comment. We do not - in general - claim to be more accurate than in-situ retrievals. We would like to compare our results to in-situ sampling.**

Reviewer Paul Connolly

This is a very well put together study, combining data from three co-located, vertically pointing radars to quantify aggregation efficiencies in the atmosphere. It is the first attempt to retrieve the ice particle size distribution from multi-frequency Doppler spectra observations. It is satisfying to see that these results, in the main, corroborate our chamber observations. The presentation is very good and there are no major issues. I recommend publication, but would like to see more information on the fall speed relations used and perhaps an assessment of how the results depend on mass - dimension relations.

**We thank the reviewer for the positive reception and interest in our results, and additionally for trying to recompute some parts of our analysis as it was helpful in highlighting which bits of information were missing from the paper.**

As I have worked on similar problems before I wanted to see whether I could reproduce the findings from the information available in the paper, to check that my interpretation is consistent with the main findings in the paper. My reasons for doing this are to demonstrate how others may interpret your data, and to check that my interpretation is correct. I present this alternate analysis, in a separate section below.

Specific Comments

Page 3, line 25 – sentence begins with "because".

**Correct. Because the subordinate clause is followed directly by the main clause, this sentence is grammatically correct.**

Page 7: Brown and Francis to convert between mass and size . The Brown and Francis (1995) relation is for ice crystals in cirrus clouds. There are more up to date mass - size relations that are published so I was curious if you have tried these, and whether a different assumption affects the results

**This is a useful suggestion. In the newly presented sensitivity analysis, a second mass-diameter relationship has been used to evaluate its importance (it turns out not to be very important). Additionally, it should be noted that the mass-diameter relationship plays no role in determining the size of the particle from the measured sDWR (only the number), and that the exponent in the mass-size relationship is shown to be consistent with the Brown-Francis value (1.9) for this case based on the fractal dimension calculation of Stein et al. (2015).**

Page 8: velocity power law – you don't give examples of the fit parameters here, which makes it more difficult for others to understand your data. I wonder if you could give some example figures, or statistics of the fit parameters (the a and b coefficients).

**Thanks for the idea. We calculated the mean and standard deviations of the a and b values (in our paper called c and d), and included these in figure 10.**

Figure 4: convincing plot. Just a comment: I am surprised that the spectral reflectivity of the middle plot and bottom plot extends to just above 1.5 m/s , whereas the size distributions are much broader for the lower layer. Is this because the larger particles in the lower layer are less dense so that their fall speed saturates with increasing particle size?

**Yes, this is an interesting aspect of this case, and already partly commented on in the text – that the sDWR of the particles around 1.5 m/s increases, but the fall velocity doesn't really change. As**

**you suggest, this could be explained by a change in density, or an insensitivity of the fall velocity to aggregation at this particular size. We do not have any data to do anything other than speculate about this.**

Alternate Analysis

**Without responding point by point to your alternate analysis , we wanted to thank you for this analysis as it 1) helped highlight which parts were missing in the paper to enable it to be understood and reproduceable and 2) brought to our attention the importance of the $I_1$ term in the Mitchell (1988) equation, as well as the mass-size and velocity-size assumptions in that calculation. We have taken your comments on board and revised the manuscript adding the relevant details. This has given us more confidence that our results are robust and has improved the quality of the paper and made our arguments more convincing. A short summary of these is below:**

- **In terms of calculating the mass flux, the velocity-size relationship is not used. Instead the Doppler velocity from the radar is used directly (details added to the paper). The snow flux and number flux (for particles D>0.75mm) is now shown in Figure 7. The values you estimated were close to ours and can be seen in the profile in Figure 7c.**
- **The velocity-size relationship is an important contributor to the $I_1$ term of Mitchell (1988). Your analysis revealed that $I_1$ is very sensitive to the velocity-size relationship used, and consequently a large uncertainty in the estimated aggregation efficiency evident. This has been brought out in the discussion in the text of the paper.**
- **Statistics of the velocity-size relationship power law fit performed have been calculated and added to the paper in the form of figure 10 and additional discussion has been added to the end of section 6 where we have through in detail about the sensitivity of our results to the various parameters input to the equation and their importance (e.g. a, b, c, d).**

Validation : in order to better understand figure 4 I thought I would do a consistency check. I digitised your data from plots in figure 4 c , f , and i First I wanted to calculate the mass flux, to see if this was roughly in - line with that expected and to see whether it was approximately conserved between levels. As you are aware the mass flux should be conserved in diffusional growth is not important. I used the Brown and Francis (1995) relation to convert particle diameter to mass (as you have done)

And, as you have not given the coefficients for the velocity size relation, I have used a fall speed relation from Wang and Chang (1993)

My analysis is shown in Figure 1 . I have calculated the mass flux at the top, middle, and bottom of the cloud presented in your figure 4 . The values I have calculated are as follows top middle bott om Mass flux 10 - 4 (kg m - 2 s - 1)

. Analysis of your figure 4.

Data points are taken from your figure 4, lines are exponential fits. Text shows the calculated mass flux. Colours are as follows: red (top of cloud); green (middle); blue (bottom of cloud).

We should expect that the mass flux increases if the particles grow by vapour diffusion, or decreases if the particles evaporate. If vapour diffusion is not important we should observe that the mass flux is conserved. Here we see approximately a factor of 1.7 reduction in the mass flux in the middle of the cloud. I suspect that these numbers are within the expected retrieval errors (or errors in mass - dimension / velocity‐dimension relations, but it would b e useful if you could comment on this. The fact that I have used a velocity‐dimension power law that is not based on your observation may also

be responsible for this too: another reason why it would be helpful to see your velocity - size relations. N ext I thought I would try the analysis of Mitchell (1988) to attempt to calculate the aggregation efficiency. The relevant equation is equation 16 in Mitchell (1988). which can be rearranged for E a , the aggregation efficiency. Here, $\beta$ =1.9, b=0.33, a=6.96, $\alpha$ =0.0185 (SI units); $\lambda$ is the slope of the size - distribution; $\chi$ f is the mass flux (the mass falling through an area per second); $\Gamma$ is the gamma function and I 1 is a definite integral to be calculated (see Ferrier et al 1994)

From the data in Figure 1 I was able to estimate !" !" to be 7.9 (SI units); $\lambda$ =6.42e3; and $\chi$ f =2.39e - 4 (SI units) are based on values in the middle of the cloud. I calculated the integral, I 1 , as 37.89 – code can be provided on request – feel free to contact me. From these numbers , and rearranging Mitchell's equation above, one can estimate Ea to be equal to approximately 0.4. This number is not too far from what you have used, but it would be useful to understand where the differences arise – I think your estimate is a little higher . For instance on page 12 you say you also use Mitchell (1988); hence, I wondered whether you could go through the calculation in more detail. I suspect this is due to the power laws used for velocity – size , but it may also be due to error s in fitting slope and intercept parameters to the data for instance. I was not sure whether you had taken into account diffusional growth either. Taking into account diffusional growth with increase the slope, so the aggregation efficiency will need to be higher than I have calculated to lead to the observed reduction in the slope. Additionally my estimate of Ea=0.4 assumes the mass flux in the middle of the cloud to be 2.4e - 4, which is low compared to the top and bottom. If I use the higher mass flux 4.8e - 4 (the value I calculated from your data at cloud base ) , in the calculation, the corresponding Ea is approximately 0.2. In addition I thought I would try and reproduce a plot similar to your figure 5c. My Figure 2 shows these simulations using aggregation efficiencies of 1, 0.4 and 0.2. The finding here is that lower values of the aggregation efficiency yield lambda values closer to your observations at the 4 km level. Again I think the reason for this discrepancy may be because my calculations have used a terminal fall speed power law that does not match the observations for small crystals. Since the calculations appear to be quite sensitive to the terminal fall speed relation it would be really useful if you could present the measured fall speed (and regression coefficients) you have used.

Figure 2 .

Model simulation using the initial conditions taken from the top of the cloud in figure 4, using different values of the aggregation coefficient.

Final word – I strongly support the statement about sizing particles down to 0.3mm, which would allow you to probe earlier stages of aggregation.

---

## Author Response (AR2)

We thank each of the reviewers for their helpful comments and criticism. We have modified the paper taking into account all of these points and feel that the revised paper is significantly improved and that our results are now presented more clearly and the findings are more robust. Here is a brief overview of the key changes made to the paper.

- Rewritten paper to be more focussed on the meteorologically interesting aspects of this case.
- Improved the clarity of the description of the new retrieval technique
- Inclusion of new equations defining the aggregation efficiency and the expected change to the lambda (slope of particle size distribution) with height due to aggregation from Mitchell (1988)
- Added more information about the assumptions made, their validity and a comprehensive sensitivity test to these assumptions (new section 6.1)
- New figures showing:
  - the relationship between diameter and DWR for the Westbrook (2006,2008)
     scattering model, and for particles from Leinonen and Moisseev (2015) (figure 1)
  - Snow flux and number flux as derived from our retrieval (figure 7)
  - Sensitivity of the particle size distribution to assumptions in the retrieval (figure 8)
  - Sensitivity of the change of lambda (slope of particle size distribution) to assumptions in the retrieval (figure 9)
  - Information about the statistical properties of the velocity-diameter power law fits made in the retrieval (figure 10)

We believe that the improved clarity and additional information now provided in the paper make our results more convincing.

Specific replies to the individual reviewers comments are below. Specific changes related to the reviewers' comments are in blue. Note that the page numbers refer to the track-changes version.

Reviewer 1

**General Comments**

The paper focuses on the identification of a rapid aggregation layer within an ice cloud. In order to do so, an innovative algorithm for the retrieval of snow particle size distribution (PSD) is developed. The algorithm leverages on the synergies of multi-frequency and Doppler observations from vertically pointing radar systems. The retrieved evolution of snowflake sizes is connected to microphysical processes through a modeling approach. It is concluded that neither depositional growth nor riming can explain alone the rapid increase in snow size and aggregation must play a major role, moreover, the expected sticking efficiency must be larger than what was previously published in dedicated laboratory experiments.

The paper puts emphasis on the properties of the rapid aggregation layer and in particular to the value of the aggregation efficiency (Eagg). This would entitle the paper to be published on ACP given the importance of this process in ice clouds. However I am not sure if the reported conclusions are sufficiently supported by scientific evidence. In particular, I am concerned about the number of strict assumptions that have been made throughout the text, the lack of independent validation of the results and the very short duration of the single event selected to support the conclusions about Eagg

value. By contrast the paper propose a very interesting and innovative way to use vertically pointing radar to retrieve the properties of particle size distributions. As best of my knowledge, this is the first retrieval of the size-resolved PSD using radars, which would allow to drop the assumptions about PSD shape that are necessary in bulk approaches. The presented methodology deserves a much more detailed description than the one presented in the text and a profound analysis of the sensitivity of the method to the various assumptions that have been made. After such revisions, I would definitely recommend to publish it, but I would suggest to consider a different journal such as AMT given the shift of the focus of the paper.

Given all of my concerns, I recommend to consider the paper to be published after a major revision.

We thank the reviewer for their comments, and have ensured that in the revised manuscript there is a clearer description of the retrieval method and a sensitivity analysis to determine the impact of uncertainties in the retrieval. Although the exact details of the particle size distribution showed variability in the sensitivity testing, the conclusion that the broadening of the size distribution is a result of aggregation appears robust. We have therefore decided to keep the paper focussed on the meteorological aspects of this event and keep the paper within ACP-scope.

Changes to paper: clearer description of retrieval method (whole of page 10) new section containing sensitivity analysis and new plots (section 6.1 and figures 8 & 9) rewritten with meteorological focus (e.g. abstract lines 1-12, better description of the case page 7, line 20 – page 9, line 4)

**Specific comments**

1) Given the centrality of the concept for the entire manuscript I would suggest to give a definition of Eagg in the introduction section. This also to prevent potential confusion, given by the non-unique nomenclature used in this field where different efficiencies might mean different things (e.g. collection efficiency, sticking efficiency). Finally, this would help understanding the reasoning behind the last paragraph of section 6 and Figure 5, where Eagg is inferred from the vertical gradient of the slope parameter of the PSD.

Thank you for the suggestion. We have added the formal definition of aggregation efficiency based on Mitchell (1988)'s aggregation kernel to the paper – see equation 1 and the supporting text.

Changes to paper: added Mitchell (1988)'s aggregation kernel (equation 1, page 3 line 12) and supporting text (page 3, lines 11-16)

2) I am not convinced by the statement about Connolly et al. (2012) at lines 31-34 of page 2. By looking at Fig. 14 in the original paper I would agree on the fact that Eagg is between 0.4 and 0.9 at -15 C because that is the confidence interval reported in the plot. For the very same reason I would say that it is between 0 and 0.5 for the other temperatures. Claiming that it is always below 0.2 might be an exaggerated statement.

We have revised the text to clarify that the "best estimate" of E\_agg is below 0.2, except at -15C.

**Changes to paper: changed text to "but the best estimate at other temperatures was below 0.2" (page 3 line 23)**

3) In section 2.1 the non-microphysical sources of differential reflectivity (DWR) are accounted for. These sources of retrieval error are compensated by making the radar reflectivity to match in the Rayleigh-scattering part of the cloud and applying the same adjustment to the whole profile (lines 4-5 of page 4). This method is strictly valid only for radar miscalibration and attenuation by radome or wet antenna; for height dependent sources of differential attenuation (e.g. atmospheric gas absorption, liquid in the cloud) this method causes an overcompensation of the higher frequency reflectivities at lower level (in particular W-band). Attenuation due to atmospheric gases depends on the density and humidity profile of the atmosphere and can adds up to 2.5 dB at the top of the cloud at midlatitude; under this condition the overcompensation caused by the radar cross-calibration at 3-4 km could be easily in the order of 1 dB. I suggest to compensate for atmospheric gases absorption profile before making the radar-cross calibration, or, at least, estimate the W-band absorption profile for the analyzed case by using either a weather model or a radiosonde profile and report it either in the paper or in supplementary material.

We understand the reviewers comment and have considered this issue. However, it should be noted that the overwhelming majority of attenuation from liquid water and gases (and therefore also differential attenuation) occurs in the lower troposphere, below the level of clouds we are analysing in this paper. Therefore, our application is analogous to that of radome attenuation (where the attenuation occurs before the first target of interest). To further support this argument, we have calculated the attenuation from water vapor as a function of height (using a nearby radiosonde profile), and determined that 85% of the attenuation occurs below cloud base. The remaining attenuation would contribute to about 0.15 dB difference of 35 and 94 GHz power. Such an offset in power translates to about 60 microns in difference of the retrieved particle size at both cloud base and cloud top. Therefore, the conclusions drawn about the rapid aggregation occurring in this cloud are not affected by the attenuation correction we have made. However, the reviewer is correct to point out that there will be some situations where this method will not work correctly and we have added a sentence to the paper stating this to discourage future researchers from blindly applying this method.

Additionally, the fact that the spectra at all 3 frequencies overlay one another nicely in the upper levels of the cloud suggest the relative calibration works well. If there were significant non-Rayleigh effects, they would be more prominent in the spectra which would show sZ94 too high for small (slow falling) particles, and sZ94 too low for large (fast falling particles). We don't see that behaviour. In order to address the reviewers concerns, we have added different possible definitions of the "Rayleigh-scattering regime" in which we match the reflectivity from the 3 radars as part of the sensitivity testing. Our findings appear robust to which range of dBZ values are chosen.

**Changes to paper:**

P6L2)

**inclusion of different possible attenuation correction ranges in new sensitivity testing section (section 6.1) clarification that the attenuation correction used here may not be suitable for all cases (P5L5-**

4) 35- and 94- GHz radars are declared to be off-zenith by 0.2 and 0.15 deg in opposite directions (lines 15-16 of page 4). This causes a contamination of the doppler signal from horizontal wind component which is then corrected by shifting the spectra by constant values (line 2 page 5). The authors

acknowledge the fact that this procedure is imperfect and consider the matching of the resulting spectra to be good in Figure 4. However it is not described how these numbers (mispointing angle and Doppler velocity correction) are obtained. I personally can hardly see how a composite relative shift of just 0.1 m/s would have affected the matching in Figure 4. The comparison in Figure 4 would have taken benefit from the inclusion of the 3 GHz spectrum which is considered to be well aligned. Also the 'goodness' of the matching is both affected by velocity and power shifts: a small residual differential attenuation would have caused the spectra to look non-aligned; given the fact that there might be still a differential attenuation issue here (see my point number 3) the matching is potentially flawed by this residual error.

I suggest to include the source of the mispointing angles and Doppler shifts numbers.

Following the reviewers suggestion, we have also added the 3 GHz spectra to the plots in Figure 4 (now Figure 5). In accordance to our reply to the previous point, the goodness of fit of all three spectra (in regions where we expect them to be identical, e.g. figure 4a (now figure5a)) provided by making these corrections to the measured Doppler velocity suggest that it is necessary and beneficial. Contrary to the reviewers expectations, a mis-alignment of the spectra by even 0.02 m/s can show substantial differences in the retrieved particle size distribution. This difference is because the spectra is relatively "steep", sZ changing rapidly as a function of velocity. Hence, slightly misaligned spectra result in artificially enhanced sDWR values, either for small velocities, or for large velocities. A relative offset of 0.1 m/s renders the retrieval useless – see the retrieved size spectra when such an uncertainty is added in the new sensitivity analysis. The importance of the velocity offset for the retrieval has been clarified in the revised paper.

A description of how the offsets and pointing angle errors were calculated was not included in the original paper because the cause of the offsets is not of practical significance, but the fact that we have corrected the data to account for them is important. For completeness we have added a short description to the paper and also here:

The velocity differences as a function of height were determined be assessing the mean Doppler velocities from the three radars individually. The correlation of these offsets with the atmospheric wind profile (determined from ECMWF forecast fields) enabled an estimation of the pointing angle errors. We include these values in the paper only to highlight to future researchers that rather small pointing angle errors can significantly affect the retrievals and therefore care should be taken to ensure that the radars are pointed as accurately as possible.

The goodness of matching of the spectra is indeed affected by both velocity and power offsets. However, these are easy to separate and correct for independently. The velocity offset can be determined through correlation of the Doppler spectra while power offsets can be determined by integrating received power across the spectra. In fact, we used bulk methods to determine the offsets (mean velocity difference between radars and total reflectivity differences – as discussed above) and these also worked well to ensure the Doppler spectra are well matched.

**Changes to paper:**

**added 3 GHz spectra to Figure 5 (page 13) the method to determine the velocity offsets as a function of height have been included (page 6 line 17-20)**

5) The PSD and v-D retrieval method is very roughly explained in pages 6 and 7. After several readings I understood that it assumed a unique relation between sDWR and the snowflake size. This relation is likely to be very specific of the assumed scattering model and mass-size relation. A plot showing this

relation for a certain number of scattering models and m-D functions will help the reader understanding the methodology applied and gives an indication of the expected uncertainty due to the related assumptions.

The reviewer is correct that this section was not sufficiently clear in the submitted version. The text has been improved to add clarity and the figure suggested has been added to the paper (new Figure 1) to highlight both the method and the characteristics of the scattering model used. There is no sensitivity of the sDWR value to the m-D relation, so this has not been included.

**Changes to paper:**

**improved description of the retrieval method (whole of page 10) inclusion of new figure 1 showing the Westbrook (2006) scattering model, and also comparison with Leinonen and Moisseev (2015) data (figure 1, page 5)**

6) Moreover, for the scattering model it is assumed Westbrook (2006, 2008a) since it has been found to closely match observation in the multifrequency space [Stein 2015]. However, the scattering model from Leinonen and Szyrmer (2015) has also been found to match the observation (unpublished on a peer review journal, but included in conference proceedings http://www.isac.cnr.it/ ~ ipwg/meetings/bologna2016/Bologna2016\_Orals/3-8\_Westbrook.pdf) It would be very interesting to see the results from this different scattering model. Being a detailed DDA model one does not have to assume the m-D relation but can simply take particles masses and sizes from the database, achieving a better consistency of the results.

While comparison with other scattering models is indeed interesting, we present plenty of evidence in this paper, and also in Stein et al. 2015, that this scattering model is suitable for this case. The Leinonen and Szyrmer (2015) scattering model is for rimed aggregates, and we have already stated in the paper that there is no evidence for riming, either in terms of the presence of supercooled liquid water or the characteristic behaviour of the pair of DWR values presented in Stein et al (2015). However, the scattering calculations in the presentation that you link to are indeed for unrimed aggregates – these have very similar characteristics to the Westbrook (2006) scattering model used. We have added points to Figure 1 to highlight the similarities of the two scattering models.

Additionally, it should be noted that we do not require a mass-size relationship to relate sDWR and particle size – the retrieval method for size is independent of the mass-size relationship. Changes to paper:

**inclusion of new figure 1 showing the Westbrook (2006) scattering model, and also comparison with Leinonen and Moisseev (2015) data (figure 1, page 5)**

7) The particles that are sampled within each Doppler bin are likely to have different sizes. Is the model considering only one particle size per Doppler bin? This is potentially a significant source of error when large particles are present. Large particles are expected to fall roughly at the same velocity for many different sizes, thus  $dv/dD \ll 1$ , by contrast the backscattering signal given by those particles is very different. Assuming that in fast-falling doppler bins (i.e. v>1m/s) there are snow particles of just one size is not a valid assumption even at for doppler systems with a very high spectral resolution.

We agree with the reviewer that the relationship between particle size and particle velocity is likely not a one-to-one monotonic function. This assumption and the limitations have been further discussed in the revised paper compared to the original version. However, as a first attempt at using such a technique to retrieve the particle size distribution without fixed assumptions about the shape of the distribution - we need to make some assumption here. It may well be that a refined approach could yield more accurate or robust results; however, this would require a-priori information on the statistical variability in V for a given D, which is poorly constrained at present, and therefore we leave that for future work.

**Changes to paper:**

inclusion of an improved description of the retrieval technique (whole of page 10)

8) Considering the number of correction, a sensitivity analysis of the algorithm with respect to the input data is essential. Assuming 1 dB uncertainty in radar reflectivity and 1 or 2 velocity bins uncertainty in the doppler spectra will already give a good indication of the robustness of the algorithm. It will be particularly interesting to see how this translates into uncertainties in the retrieved PSDs (panels c, f and i of figure 4) and the profile of fitted scale parameter lambda in figure 5.

A sensitivity analysis incorporating uncertainties of +/- 1 dB to the Doppler spectra, a shift in velocity space of up to +/- 0.04 m/s, which range of reflectivity values are used for attenuation correction and an additional mass-diameter relationship has been added. The impact of the uncertainties of the size distribution (equivalent to figure 4i) and of the vertical profile of Lambda (equivalent to figure 5) have been added. The sensitivity analysis adds confidence to our argument that aggregation is the primary driver of change to the size distribution in the lower region of the cloud. Furthermore, it shows the (negative) impact on the particle size distribution retrieval when the Doppler spectra are not suitable matched – suggesting that our matching methodology is in fact reliable (albeit imperfect).

**Changes to paper:**

completed new sensitivity analysis and added this to the paper (section 6.1) and figures 8 & 9.

9) The result of 'rapid aggregation' is obtained by comparing the relative potential of various snow growth processes, concluding that only aggregation is capable to give such rapid change in PSDs scale parameter lambda. I think that the potential given by the PSDs retrieval is here underutilized. Given the full PSD and the m-D and v-D relations one can calculate interesting bulk quantities such as particle number concentration (Nt) and ice water content (IWC) and their vertical fluxes (particle flux Nf and snow rate SR). It will be extremely interesting to see time-height plots of this quantities in connection with the newly developed mode in fig4d. This analysis would also help in the identification of the significant aggregation process. Depositional growth and riming are in fact expected to increase SR leaving Nf unchanged (unless newly nucleating particles are present). Aggregation has the distinctive effect of decreasing Nf leaving SR unchanged and this should appear in the suggested plots.

We thank the reviewer for this suggestion, which has resulted in the addition of a new figure showing the number flux and snow flux throughout the interesting part of the observed cloud. We believe that the coherence in these plots adds support that our retrieval is stable and as the revierer suggested – the vertical profile of the number and snow flux do add support to the rapid aggregation hypothesis. Although the snow flux decreases with height (presumably due to sublimation), the number flux decreases with height significantly faster.

**Changes to paper:**

New figure 7 shows the snow flux and number flux as the reviewer suggested – and shows the

**contrast between snow flux and number flux consistent with the aggregation process (figure 7, page 18)**

10) At line 27 of page 12 it is mentioned that the methodology described in Mitchell (1988) has been used to model the evolution of lambda parameter, however it is not specified the exact model used. It is surprising that the formula for the depositional growth rate from Pruppacher and Klett (1978) has been fully reported and not this. This is potentially a serious issue regarding the reproducibility of the results. Additionally, I think that a better explanation of the model used will give the reader a better understanding of the other variables that influence the PSD evolution due to aggregation such as particle sizes, velocity differences and total number concentration.

We apologise for this oversight and have now added the full equation from Mitchell (1988) to our paper. Furthermore, because of the additional analysis of the snow flux added, we decided that the constant snow-flux with height assumption was not valid, and instead used the retrieved snow flux profile in the calculation. We have maintained the constant snow-flux profile for E\_agg of 1.0 in the paper for comparison.

**Changes to paper:**

Added the full equation from Mitchell (1988) (equation 5, page 16) Changes to our calculations resulting from removing the constant snow flux assumption are now seen in figure 6c (page 17)

11) The conclusions about the value of the Eagg are supported by only a 2 minute average profile during one event. I would, at least, model the evolution of the lambda parameter for other times during the same event, or, even better, model more events.

While we understand the reviewers request for the analysis of more data, we focus in this paper on the most microphysically interesting part of the cloud. We do not claim to make any general quantification of the aggregation efficiency, but rather to say that in this instance the observations are consistent with a large aggregation efficiency and the new retrieval has helped identify this process.

We have collected radar data from several days and believe that we have seen similar events within that dataset. We have focussed on this case study as it is microphysically interesting enough to serve as an example of the new retrieval technique and the insights into cloud microphysical processes that it can provide. The analysis of further events, where similar features were observed, are underway. The analysis of these events is not ready to be included in this paper and will hopefully be published separately (although the research grant for this work has now expired).

**Technical corrections**

12) When presenting the state of the art of Doppler/multi-frequency radar retrievals at line 20-25 of page 2, I suggest to consider some recent studies like Chase et al. (2018) or Leinonen et al. (2018) in the discussion.

**Thanks. These more recent papers have been added to this discussion.**

**Changes to paper: added the suggested references (page 3, line 8)**

13) The choice of the colormap used in figures 2, 3 and 5 is particularly unfortunate. There is an apparent overlap of light-blue colors for different values that makes the interpretation of the figures

more difficult than it should be. In figure 4b there are vast areas of the cloud where I cannot say if the DWR is either +1 or -1 dB. In figure 5 the mapping from the colorplot to the profile is made even more difficult by the fact that the profile as been cut from the panel with a white line; here I also suggest to indicate the profile with a thin rectangle around instead of the white line.

Actually the white line in figure 5 is because of missing data. The profile is at 1615 UTC, which has been emphasised by adding the time to the title of panel c and as an additional label on the x-axis of panels a and b.

We have experimented with different colour scales; however, none are able to cover the full data range while enabling values to be read from the figure accurately. It is because of this that we added vertical profiles of the values at 1615 UTC to figures 3, 6 and 7.

```
Changes to paper:
added label to x-axis and panel c clearly stating the 1615 UTC time. (figures 3, 6 & 7; pages 8, 16,
17)
```

14) Figure 4 – Personally I would swap the axis in panels b, e, h. This would put velocity on x-axis, matching the concept on panels a, d and g. Also it appears that DWR is rather a function of velocity and not the opposite (see in particular panel e). That is a personal preference and I would leave to the authors the decision.

This is a good suggestion and has been changed in the new version of the figure.

Changes to paper: axes swapped on panels b, e, h of figure 5 (page 13)

**Reviewer2**

**1 Summary**

This manuscript proposes a new algorithm to retrieve the particle size distribution (PSD) from vertically pointing Doppler profilers at 3 frequencies, using the spectral dual-wavelength ratio and not the ratio of integrated reflectivity values as in previous work. This algorithm is then applied in the context of the study of a given cloud, to investigate the dominant microphysical processes taking place and explaining the measured Doppler spectra. Rapid aggregation appears to be the best candidate among various processes to explain the observed behavior.

**2 Recommendation**

The algorithm and the application for microphysical interpretation that are presented in this manuscript are innovative and relevant. A "direct" PSD estimation without any assumption about its mathematical functional form is promising and useful. But there are also a number of assumptions that are required to run this "inversion", and they are not all clearly described and discussed. It is hence difficult to understand in which framework this approach can be safely used, and the example presented in this manuscript remains rather specific. The manuscript is pleasant to read with quality illustrations. Overall, I am convinced that this manuscript presents innovative and original material that are worth publication, but after having addressed the issues listed below.

We thank the reviewer for their supportive comments. We have developed the manuscript in response to the reviewers comments (as detailed below) with a more rigorous analysis of the uncertainties and a better description of the assumptions and retrieval technique. As a result, we feel this has made the paper much more convincing and the results robust.

Changes to paper: rewritten the description of the retrieval technique (whole of page 10) added comprehensive sensitivity analysis (section 6.1 and figures 8 & 9)

**3 General comments**

1. Information about the methodological side is missing: no detailed/exhaustive description of the proposed PSD spectral retrieval algorithm is provided, making it difficult to check or reproduce for instance. I suggest the authors to add detailed description (including equations and so on) of the different steps of the algorithm.

**The algorithm is actually relatively simple, but we acknowledge that the description could have made the process clearer to the reader. We have added additional text and clarifications and believe that the revised manuscript provides a better basis for reproducing this method.**

**Changes to paper: rewritten the description of the retrieval technique (whole of page 10)**

2. The case study is too limited (40 min of a single cloud) to derive general insights beyond the demonstration that the proposed method works, at least for one cloud. I understand the difficulty to expand the analysis, but this example is too limited in itself (see below).

The focus on this case is because of the interesting, abrupt in height but consistent in time, appearance of large DWR values lower in the cloud. We do not try to claim that this is a common feature, or that the aggregation efficiencies derived are common to other clouds/cloud types. As

discussed in the paper, data on the aggregation efficiency is rare and there is a large spread in the reported values. Therefore we believe the additional insight from this paper to be valuable to the community. We present an analysis of what is occurring at this time and height in this cloud, and show how this method can be useful to investigate processes in such clouds. Further study of other clouds using the same method is underway at a preliminary stage, and we hope that this study and retrieval technique will provide a foundation to analyse the variability of aggregation efficiency in clouds in a systematic way and to evaluate how it depends on temperature, relative humidity etc.

3. From a more general point of view, I have the feeling that this manuscript "oscillates" between the two Copernicus journals AMT and ACP, between a more methodological point of view (e.g. the retrieval algorithm) and a more meteorological point of view (case study of rapid aggregation in a cloud). So in the end, the reader is somehow frustrated: on the one hand, the paper proposes a new retrieval method (AMT side), but does not provide enough description of this method for the reader to implement it; on the other hand the case study is too limited to gain any general insights into cloud microphysics (ACP side). I am fine with the authors choosing ACP, but I would strongly recommend to add more explanations about the proposed retrieval technique, as well as more discussion about the limitations and the conditions in which this approach is valid.

There is some content in this direction in the conclusion (p.15, I.7-11) but only the verticality and the beam width are discussed, not the requirements in terms of turbulence, (supercooled-)liquid water or not, the geographical representativity, etc.

Based on this comment and that from other reviewers, we have tried to focus more on the meteorological aspects within the paper, but at the same time clarifying adding some more details about the retrieval technique.

Changes to paper:

rewritten the paper to be more focussed on the meteorological aspects (throughout)

4 Specific comments

1. P.8, I.2: optimal with respect to what? Which fitting method is employed to estimate the powerlaw parameters?

The word optimal has been removed. The power-law was estimated by fitting a linear best-fit line to the logarithm of the values.

Changes to paper: removed the word "optimal" (page 10, line 28)

2. P.8, I.2: why using a power law between vertical terminal velocity and the size?

The power law has been used because 1) it is easily differentiable and 2) it is common in microphysical scaling relationships. Changes to paper:

justified use of power law relation (page 10, lines 29-31)

3. P.8, I.10: so the 3 GHz spectra are used "only" for large particles? If so, the proposed approach is essentially dual-frequency. Should the title be adapted?

The 3-GHz is essential for the attenuation correction of the radars (because it is used to identify the Rayleigh-scattering part of the cloud and provide a reference). It is used in the sizing of

particles larger than 2.2 mm (which can be done for both 35 and 94 GHz, and should provide the same answer). It is furthermore useful to help identify the correct scattering model to use – as done in Stein et al (2015). Therefore, although some aspects only employ dual-frequency techniques, the complete retrieval is dependent on all three radars. The 3GHz spectra has additionally been added to Figure 5 to enable comparison between all three radars.

**Changes to paper:**

added the 3 GHz spectra to figure 5

added examples of other uses of the 3 GHz data, for example in determining the correct scattering model (page 10, line 5-10) and the exponent for the mass-size relationship (page 10, line 14-16)

4. P.10, I.26-27: what are the plausible mechanisms to explain the generation of these new ice particles? Maybe it was mentioned somewhere but if so, I missed it.

**We have not speculated on the generation mechanism because we have no data that will help determine or rule out any mechanism.**

5. P.11, l.25-27: is a SNR threshold applied prior to run the retrieval, in order to filter out the noisy values?

Yes, noisy data points are filtered out and details have been added to the text.

Changes to paper: clarification of the data processing section (specifically page 7, lines 10-15)

**Reviewer 3**

The authors present a method to quantify the aggregation process and retrieve the ice particle size distribution using three co-located radars. They showed that aggregation causes a rapid (less than 10 minutes) growth of ice particles from 0.75 mm to 5 mm in maximum size. They speculate that the dendrites dominate at -15 C with large aggregation efficiency (approximated to be near unity). Although the results are important and the manuscript is interesting, there are multiple issues that have to be addressed before the manuscript can be accepted for publication. My suggestions are explained below.

**General comments:**

- How do you distinguish between the ice particles and water drops? In pg 5, In 21, you said that your case is an ice cloud. Elsewhere you mentioned that there was no water drop in the cloud. However, a mixed-phase cloud is probable in this temperature range. Fig. 3 shows that the temperature in the presence of cloud ranges from 0 to -40 C. Between -38 and 0 C, super-cooled water drops co-exist with ice particles (Rosenfeld and Woodley, 2000), and there is a great chance of water contamination.

It is important to address this, and explain how you detect water drops and exclude them. Alternatively, is it possible to quantify the ratio of liquid water content to ice water content?

Mixed-phase clouds are possible in this temperature range; however, we are confident that the liquid water content in this cloud is negligible. Firstly, the microwave radiometer instrument does not detect any significant liquid water. Second, analysis of the radar Doppler spectra does not show any evidence of low reflectivity drops at small fall velocities (although they could be too small to be detectable). Third, the evidence of the pairs of DWR-values shown in Stein et al. (2015) are consistent with aggregates, and inconsistent with rimed particles – which suggests that there isn't a lot of supercooled water present.

**Changes to paper:**

**These clarifications are all included in the revised paper (page 15, line 24-29)**

- There is no comparison between your retrieval and direct measurements of size spectra, because there was no in-situ measurement available for your case. It is true that disagreements exist between various in-situ probes (see also Fig. 6 in Cotton et al., 2010), but still it is not certain if your retrieved size spectra would be more accurate. It would be good to cite any study that compared retrieved size spectra with direct observations. In any case, such caveat (no comparison between your retrieval and in-situ measurements) should be explained in the manuscript, and should be mentioned in the abstract and conclusions.

**We do not try to argue that our method is necessarily more accurate than in-situ measurements. However, the advantage is that we can make continuous measurements and multiple heights simultaneously and see the evolution of the size distribution. It is unfortunate that there is no insitu data available, and such a comparison is part of the planned future work.**

- The Brown and Francis (1995) mass-size relation has an important issue: it's not realistic for size smaller than 100 microns, since it gives ice particle mass larger than that of a sphere. See Erfani and Mitchell (2016) and their Fig. 1. I understand that you do not detect particle smaller than 0.75 mm, but it is important to address this issue for the readers who might use Brown and Francis mass-size relation. In addition, the readers will become aware of the more recent mass-size relations.

We agree that the Brown and Francis (1995) mass-diameter relationship is not physical for small sizes. However, such a failing does not affect our retrieval because it is not possible to reliably size any particles smaller than 0.75 mm, and certainly not down to the 100 micron scale. We also note that Brown and Francis do address this issue in their original paper and for these reasons it seems unnecessary to include a repetition of that information in our paper.

As part of the newly added sensitivity analysis, uncertainty to the mass-diameter relationship is estimated. We also have evidence from the Stein et al. (2015) paper that the exponent (1.9) used in the Brown and Francis mass-diameter is consistent with the observations on this day.

**Changes to paper:**

mass-diameter relationship uncertainty included as part of the sensitivity testing (section 6.1 and figures 8 & 9) additional text clarifying that aggregation calculation is independent of mass-size relationship (section 6.1) clarification that the mass-diameter relationship from Brown & Francis is used for this case

because it is a good fit to the data, but that it may not be applicable for all cases (page 10, line 11-16)

- Your radar is unable to detect particles smaller than 0.75 mm. This means that your retrieved data is not able to approximate the vast majority of particle number density or dN/dD (because small particles dominate the number concentration or N; again see Fig. 6 in Cotton et al., 2010). How does that affect your calculations? Since the calculation of number concentration is an important part of your paper, you should highlight this limitation (no detection for size less than 0.75 mm) and its consequences in the abstract and conclusions.

Perhaps it was not clearly written in the paper, but it is incorrect to say that the radars cannot detect particles smaller than 0.75 mm diameter, but rather that their size cannot be determined reliably. We have attempted to clarify any possible misunderstanding in the revised paper, including a statement in the abstract that the size distribution can only be estimated for particles >0.75mm in diameter. The radars detect all particles of all sizes (assuming that there is enough total signal to differentiate that from background noise). While you are correct that the retrieval of the number concentration is important for the size distribution – we only attempt to retrieve the size distribution where sufficiently large particles exist. We could extrapolate back to small sizes from the fitted size distribution to determine an approximate number of small particles, but we have no need to do this for our study.

Changes to paper: change to the abstract clarifying that only the size distribution is only estimated for particles >0.75 mm (page 1, line 8)

Specific comments:

- abstract, In 5: Did you calculate the mean size change by aggregation?

Note that we are not referring to the mean size here (an advantage of our spectral technique). We argue that almost all of the size change is due to aggregation. We explain in the next sentence in the paper that the increase in size is shown to be consistent with aggregation when Eagg=0.7.

- abstract, In 11: Any evidence to support this? I understand that this is suggested based on previous studies. If yes, it should be mentioned explicitly: "Based on previous study, we suggest ..."

**There is no direct evidence of this; however, such a process would be consistent with large aggregation efficiencies.**

- pg 2, ln 7: By "cloud microphysical properties", do you mean individual ice particle properties such particle size or mass?

Yes, but not only individual ice particles properties, also bulk properties such as ice water content or equivalent properties for liquid water.

- pg 2, ln 14: Please add at least one example (with a reference) on how different size spectra affect the relative importance of microphysical processes.

We have added examples of why vapor deposition, riming and aggregation are affected by the particle size distribution. Details have been added to the text. Furthermore we have added numerous references to the importance of particle size distributions on correctly simulating different cloud types (page 4, line 3-7).

Changes to paper:

Examples of why vapor deposition, riming and aggregation are affected by the particle size distribution have been added (page 2, line 22-27) The importance of the particle size distribution for different microphysical processes from several references has been added (page 4, line 3-7)

- pg 2, ln 15: Please add a reference.

**The sentence reads "Another important application is to provide observations with which numerical models can be evaluated and their parameterizations improved." We do not think any reference is necessary here.**

- pg 2, last paragraph: It is good to cite Keith and Saunders (1989), since they performed experiments and measured the aggregation efficiency for various shapes and sizes. They showed that the aggregation efficiency ranges between 0.3 and 0.85 for planar snow crystals depending on the particle size.

Thanks, we were previously unaware of this paper and we have now added this reference to the discussion

**Changes to paper: added discussion of Keith and Saunders paper (page 3, line 23-25)**

- pg 3, Section 2: Please add proper references for each radar and for the near-field correction method. Overall, this section doesn't have sufficient citations and I can see only 2 references in the whole section.

We have added the references for the three radars. The near-field correction was derived empirically by comparison of the 3 and 35 GHz radar profiles in low-reflectivity (Rayleigh scattering) ice clouds and is discussed briefly in Stein et al. (2015). This section is describing the data that we collected, and therefore additional citations would not be relevant here.

> Changes to paper: added references for the 3 radars (page 4, line 13-15) added reference for near-field correction (page 4, line 26)

- pg 4, paragraph starting at ln 14: Have you tried to correct the direction of 2 radars and make a few measurements, and then compare with the previous measurements?

The errors in the pointing angles were only identified through the analysis in this paper. We could therefore not correct the pointing angles in time to observe this cloud.

Changes to paper: clarification as to how the pointing angle errors were identified (page 6, line 14-20)

- Table 1: Right now, it is not mentioned anywhere in the manuscript.

This has now been corrected.

Changes to paper: reference to Table 1 added to the appropriate discussion (page 6, line 10-11)

- pg 5, ln 26: Are these temperatures measured by radiosonde?

The temperatures were from a nearby radiosonde and also from ECMWF forecasts.

- pg 5, ln 28, Change to "Figure 2b".

The reference to "Figure 2" is correct since we refer to both the time series of reflectivity and DWR, which are shown in separate panels.

- pg 5, ln 30: Was the Westbrook model initialized for the same cloud?

The scattering model predicts the observed radar reflectivity based on characteristics of the ice aggregates generated by an idealised model of the aggregation process. It is therefore a statistical relationship and does not use any measurements of the cloud on this day. However, as shown by Stein et al. (2015), there is good agreement between the expected behaviour of the DWR pairs and that predicted by the Westbrook scattering model.

**Changes to paper:**

added figure 1 illustrating the D vs DWR relationships from the Westbrook scattering model. clarified that the use of the Westbrook scattering model for this case is supported by findings from Stein et al (2015) (page 10, line 5-10)

- Fig. 2b an 2c: The explanation of Fig. 2b in the manuscript is not enough. What is the physical interpretation of such difference between the two radars. Also, the explanation of Fig. 2c is missing in the manuscript.

Further explanation of what is now figure 3a,b has been added, explaining the quantities further and adding brief physical interpretation. Panel c is now explicitly referenced within this discussion.

Changes to paper: added better description of the case including better description of all panels in figure 3 (page 7, line 20 – page 8, line 4)

- Fig.3 and 4: You explained Fig. 4 in the manuscript earlier than Fig. 3, so please switch these figures.

**The discussion of figure 3 (now figure 4) has been moved earlier**

Changes to paper: figure 4 is now discussed before figure 5 (page 8, line 5 – page 9, line 4) - pg 6, ln 6: Briefly define the scattering model. Also, do you mean individual ice particle or a bulk property such as mean size or median size?

**The discussion has been complimented by the new Figure 1 showing how the DWR values change as a function of particle diameter for the scattering model. The diameter is that of an individual particle.**

**Changes to paper:**

**figure 1 and discussion thereof have been added (page 8, line 1-3 & page 10, line 5-7)**

- pg 7, ln 2: See the general comment regarding mass-size relation. It would be good to cite Erfani and Mitchell (2016) since they explained recent mass-size relations.

The general comment was noted. As a result of reviewers comments, the sensitivity to the choice of mass-diameter relationship is now explicitly included within the sensitivity testing. Erfani and Mitchell (2016) provide more accurate but more complex mass-diameter relationships, whereas the estimation of the aggregation efficiency from Mitchell (1988) requires a mass-diameter relationship of the form m=aD^b.

**Changes to paper: mass-diameter relationship is included in the new sensitivity analysis section (section 6.1 and figures 8 & 9)**

- pg 7, ln 1-4: The steps 2 and 3 need to include the relationships you used to relate various variables.

**We have clarified this section to address these and other concerns.**

**Changes to paper: clarifications to the whole of the retrieval process (whole of page 10)**

- Fig. 4: When the x-axis says "particle diameter", do you mean the maximum size of each particle, or did you calculate the sphere-equivalent diameter? Moreover, the explanation of panels b-e-h is missing in the manuscript.

**The particle diameter is the maximum dimension, following Westbrook et al. (2006) Discussion of panels b-e-h added.**

**Changes to paper:**

**discussion of panels b, e, h have been added and clarified (page 12, lines 9, 14, 6)**

- pg 10, ln 15-16: When particle sizes grow, but their fall speed does not increases, this is a sign of branching and aggregation rather than riming. See Locatelli and Hobbs (1974).

**At this point in the manuscript we do not make any assertions about the processes involved.**

- pg 10, ln 17-24: Combine all these lines into one paragraph.

**Done**

- pg 10, ln 32: Doppler spectra is not bi-modal in Fig. 4a. Do you mean Fig. 4b?

**Yes, corrected. Thanks.**

- pg 10, ln 33: Why are such small particles a result of nucleation and not growth by vapor deposition, or a secondary ice production (such as fragmentation of ice particles)? Elsewhere you assumed the

small particles in the bi-modal spectra are the result of vapor deposition. Any evidence on the mechanism responsible for the increase in small particles?

**You are correct, we do not know how these particles have formed, possibly through nucleation, shattering or other processes.**

**Changes to paper: "Nucleated" has been replaced by "formed" in the text.**

- pg 11, ln 1: You say the aggregation causes ice particles to grow larger and fall faster, but aggregate fall speed does not grow by size. See Locatelli and Hobbs (1974) and their Fig. 12. They also provide fall speed-size relations for various ice particle shapes (including dendrites and aggregates). It's good to cite this paper, and also it would be great if you fit their relation to your data and calculate the R-squared.

**This is not what the sentence states. It says that the sDWR increases for the larger particles (which are also the fastest falling particles), but not that they necessarily fall faster.**

- pg 11, ln 3-4: This can be a sign of aggregation.

**This is the argument that we make later in the paper. Here we only present the evidence.**

- pg 11, ln 11: This is an exponential function. Moreover, I assume D and dN/dD are known in this equation. How did you calculate NO? It is important to explain this in the paper. It seems that the value of slope is dependent on the calculation of NO. Furthermore, do you use such distribution to relate size to radar reflectivity? Your size spectra do not include small particles. Since the number of small particles contributes significantly to the number concentration, how did this affect your calculations?

**Both N0 and Lambda are determined by fitting a straight line to the size distribution data (D and log10 (dN/dD)).**

**Such a fixed size distribution shape is not part of the retrieval, which is one advantage of our method.**

- pg 11, ln 13: This is a qualitative comparison. Have you looked at the difference between Fig. 5a and 5b? From Fig. 5c, it seems the agreement between the 2 slopes is not excellent. Note that this is a logarithmic axis and I can see the red line can be larger by a factor of 1.5.

No, we have not performed a quantitative comparison. However, there good agreement in the values and patterns shown by the two methods. The disagreement at lower heights possibly highlights a weakness in the DWR method, where a fixed size distribution shape is required. We have no expectation that these two (nearly-independent) methods would give such a good agreement on the trend of lambda with height. The absolute values are not particularly interesting for our purposes.

- pg 12, ln 10-12: How did you calculate F (ventilation coefficient) and K (thermal conductivity)?

Suitable values for the appropriate temperature range were chosen. (K=0.024, Rogers & Yau, chapter 7; F=0.65 + 0.44\*0.6^0.33 Re^0.5; Re = rho \* D \* V(D) / dynamic\_viscosity; dynamic\_viscosity is in the range 1.512E-5 ... 1.862E-5, depending on temperature.

**Changes to paper: A clearer description has been added to the text (page 15, line 16-17)**

- pg 12, ln 26-27: The vapor deposition and riming do not change the total number of ice particles (N), but they do change the number of ice particles within each size bin (dN/dD). Please clarify that the rate of change in size is not the same for all sizes. As an example for riming: riming collision efficiency is a strong function of ice particle size, so larger ice particles would grow faster due to riming. See Wang and Ji (2000) and their Fig. 7; might be good to cite this paper.

We have chosen to remove this statement from the paper as it is not key to the argument that aggregation is the main process acting in this cloud and adds unnecessary confusion. We present sufficient evidence in the rest of the paper that aggregation is important and that deposition and riming are not the key processes involved.

Changes to paper: removed sentence (page 16, line 1-2)

- pg 12, ln 27-28: Please refer to the proper equation number in Mitchell (1988).

The full equation is now reproduced in the paper.

Changes to paper: added equation 5 (page 16, line 3)

- pg 14, last paragraph: This statement is suited for the Introduction and can be moved near the end of Introduction as the motivation for your study.

**Agreed. This paragraph of text has been moved**

**Changes to paper: moved paragraph (page 21, line 15 – page 22, line 4 moved to page 4, line 2-7)**

- pg 15, first paragraph: See my general comments on the lack of comparison with in-situ measurements; I agree the issues exists in directly measuring the particle size and concentration, but still it is unclear how your method has better accuracy. In addition, please cite Cotton et al. (2010) when explaining the disagreements in the in-situ measurements of ice particles.

Again, see the above comment. We do not - in general - claim to be more accurate than in-situ retrievals. We would like to compare our results to in-situ sampling.

This is a very well put together study, combining data from three co-located, vertically pointing radars to quantify aggregation efficiencies in the atmosphere. It is the first attempt to retrieve the ice particle size distribution from multi-frequency Doppler spectra observations. It is satisfying to see that these results, in the main, corroborate our chamber observations. The presentation is very good and there are no major issues. I recommend publication, but would like to see more information on the fall speed relations used and perhaps an assessment of how the results depend on mass - dimension relations.

**We thank the reviewer for the positive reception and interest in our results, and additionally for trying to recompute some parts of our analysis as it was helpful in highlighting which bits of information were missing from the paper.**

As I have worked on similar problems before I wanted to see whether I could reproduce the findings from the information available in the paper, to check that my interpretation is consistent with the main findings in the paper. My reasons for doing this are to demonstrate how others may interpret your data, and to check that my interpretation is correct. I present this alternate analysis, in a separate section below.

Specific Comments

Page 3, line 25 – sentence begins with "because".

**Correct. Because the subordinate clause is followed directly by the main clause, this sentence is grammatically correct.**

Page 7: Brown and Francis to convert between mass and size . The Brown and Francis (1995) relation is for ice crystals in cirrus clouds. There are more up to date mass - size relations that are published so I was curious if you have tried these, and whether a different assumption affects the results

This is a useful suggestion. In the newly presented sensitivity analysis, a second mass-diameter relationship has been used to evaluate its importance (it turns out not to be very important). Additionally, it should be noted that the mass-diameter relationship plays no role in determining the size of the particle from the measured sDWR (only the number), and that the exponent in the mass-size relationship is shown to be consistent with the Brown-Francis value (1.9) for this case based on the fractal dimension calculation of Stein et al. (2015).

**Changes to paper:**

**addition of the sensitivity testing section including a different mass-diameter relationship (section 6.1 and figures 8 & 9)**

Page 8: velocity power law – you don't give examples of the fit parameters here, which makes it more difficult for others to understand your data. I wonder if you could give some example figures, or statistics of the fit parameters (the a and b coefficients).

Thanks for the idea. We calculated the mean and standard deviations of the a and b values (in our paper called c and d), and included these in figure 10.

Changes to paper: added new figure 10 and discussion of the figure to give an indication of the values of c and d from the power-law fits (figure: page 21)

**Additionally, we included text describing how uncertainty in c and d follow through to uncertainty in the aggregation efficiency calculated, and which values are constrained by other measurements (page 19, line 12-18)**

Figure 4: convincing plot. Just a comment: I am surprised that the spectral reflectivity of the middle plot and bottom plot extends to just above 1.5 m/s, whereas the size distributions are much broader for the lower layer. Is this because the larger particles in the lower layer are less dense so that their fall speed saturates with increasing particle size?

Yes, this is an interesting aspect of this case, and already partly commented on in the text – that the sDWR of the particles around 1.5 m/s increases, but the fall velocity doesn't really change. As you suggest, this could be explained by a change in density, or an insensitivity of the fall velocity to aggregation at this particular size. We do not have any data to do anything other than speculate about this.

**Alternate Analysis**

Without responding point by point to your alternate analysis , we wanted to thank you for this analysis as it 1) helped highlight which parts were missing in the paper to enable it to be understood and reproduceable and 2) brought to our attention the importance of the I1 term in the Mitchell (1988) equation, as well as the mass-size and velocity-size assumptions in that calculation. We have taken your comments on board and revised the manuscript adding the relevant details. This has given us more confidence that our results are robust and has improved the quality of the paper and made our arguments more convincing.

**Changes to paper:**

In terms of calculating the mass flux, the velocity-size relationship is not used. Instead the Doppler velocity from the radar is used directly (details added to the paper, page 16, line 11-15). The snow flux and number flux (for particles D>0.75mm) is now shown in Figure 7. The values you estimated were close to ours and can be seen in the profile in Figure 7c.
 The velocity-size relationship is an important contributor to the l1 term of Mitchell (1988). Your analysis revealed that l1 is very sensitive to the velocity-size relationship used, and consequently a large uncertainty in the estimated aggregation efficiency evident. This has been brought out in the discussion in the text of the paper. (page 19, lines 3-9)
 Statistics of the velocity-size relationship power law fit performed have been calculated and added to the paper in the form of figure 10 and additional discussion has been added to the end of section 6 (page 19, lines 3-9) where we have thought in detail about the sensitivity of our results to the various parameters input to the equation and their importance (e.g. a, b, c, d).

Validation : in order to better understand figure 4 I thought I would do a consistency check. I digitised your data from plots in figure 4 c , f , and i First I wanted to calculate the mass flux, to see if this was roughly in - line with that expected and to see whether it was approximately conserved between levels. As you are aware the mass flux should be conserved in diffusional growth is not important. I used the Brown and Francis (1995) relation to convert particle diameter to mass (as you have done)

And, as you have not given the coefficients for the velocity size relation, I have used a fall speed relation from Wang and Chang (1993)

My analysis is shown in Figure 1. I have calculated the mass flux at the top, middle, and bottom of the cloud presented in your figure 4. The values I have calculated are as follows top middle bott om Mass flux 10 - 4 (kg m - 2 s - 1)

. Analysis of your figure 4.

Data points are taken from your figure 4, lines are exponential fits. Text shows the calculated mass flux. Colours are as follows: red (top of cloud); green (middle); blue (bottom of cloud).

We should expect that the mass flux increases if the particles grow by vapour diffusion, or decreases if the particles evaporate. If vapour diffusion is not important we should observe that the mass flux is conserved. Here we see approximately a factor of 1.7 reduction in the mass flux in the middle of the cloud. I suspect that these numbers are within the expected retrieval errors (or errors in mass - dimension / velocity - dimension relations, but it would b e useful if you could comment on this. The fact that I have used a velocity - dimension power law that is not based on your observation may also be responsible for this too: another reason why it would be helpful to see your velocity - size relations. N ext I thought I would try the analysis of Mitchell (1988) to attempt to calculate the aggregation efficiency. The relevant equation is equation 16 in Mitchell (1988). which can be rearranged for E a , the aggregation efficiency. Here,  $\beta = 1.9$ , b=0.33, a=6.96,  $\alpha = 0.0185$  (SI units);  $\lambda$  is the slope of the size - distribution;  $\chi$  f is the mass flux (the mass falling through an area per second);  $\Gamma$  is the gamma function and I 1 is a definite integral to be calculated (see Ferrier et al 1994)

From the data in Figure 1 I was able to estimate !" !" to be 7.9 (SI units);  $\lambda$  =6.42e3; and  $\chi$  f =2.39e -4 (SI units) are based on values in the middle of the cloud. I calculated the integral, I 1, as 37.89 - code can be provided on request - feel free to contact me. From these numbers, and rearranging Mitchell's equation above, one can estimate Ea to be equal to approximately 0.4. This number is not too far from what you have used, but it would be useful to understand where the differences arise - I think your estimate is a little higher . For instance on page 12 you say you also use Mitchell (1988); hence, I wondered whether you could go through the calculation in more detail. I suspect this is due to the power laws used for velocity - size, but it may also be due to error s in fitting slope and intercept parameters to the data for instance. I was not sure whether you had taken into account diffusional growth either. Taking into account diffusional growth with increase the slope, so the aggregation efficiency will need to be higher than I have calculated to lead to the observed reduction in the slope. Additionally my estimate of Ea=0.4 assumes the mass flux in the middle of the cloud to be 2.4e - 4, which is low compared to the top and bottom. If I use the higher mass flux 4.8e - 4 (the value I calculated from your data at cloud base), in the calculation, the corresponding Ea is approximately 0.2. In addition I thought I would try and reproduce a plot similar to your figure 5c. My Figure 2 shows these simulations using aggregation efficiencies of 1, 0.4 and 0.2. The finding here is that lower values of the aggregation efficiency yield lambda values closer to your observations at the 4 km level. Again I think the reason for this discrepancy may be because my calculations have used a terminal fall speed power law that does not match the observations for small crystals. Since the calculations appear to be quite sensitive to the terminal fall speed relation it would be really useful if you could present the measured fall speed (and regression coefficients) you have used.

Figure 2.

Model simulation using the initial conditions taken from the top of the cloud in figure 4, using different values of the aggregation coefficient.

Final word - I strongly support the statement about sizing particles down to 0.3mm, which would allow you to probe earlier stages of aggregation.

**Rapid ice aggregation process revealed through triple-wavelength Doppler spectra radar analysis**

Andrew I. Barrett1,2, Christopher D. Westbrook1, John C. Nicol1, and Thorwald H. M. Stein1 1Department of Meteorology, University of Reading, Reading, RG6 6BB, UK 2Institute for Meteorology and Climate Research, Karlsruhe Institute of Technology, Karlsruhe, 76131, Germany **Correspondence:** Andrew Barrett (andrew.barrett@kit.edu)

**Abstract.**

Rapid-We have identified a region of an ice cloud where a sharp transition of dual-wavelength ratio occurs at a fixed-height for longer than 20 minutes. In this paper we provide evidence that rapid aggregation of ice particles has been identified by combining data from three co-located, vertically-pointing radars operating at different frequencies. A new technique has been

- 5 developed that uses occurred in this region creating large particles. This evidence comes from triple-wavelength Doppler spectra radar data that were fortuitously being collected. Through quantitative comparison of the Doppler spectra from these radars to retrieve the vertical profile of the three radars we are able to estimate the ice particle size distribution (of particles larger than 0.75 mm) at different heights in the cloud. This allows us to investigate the evolution of the ice particle size distribution. The distributions, distribution and determine whether the evolution is consistent with aggregation, riming or vapour deposition. The
- 10 newly-developed method allows us to isolate the signal from the larger (non-Rayleigh scattering) particles in the distribution. Therefore, a particle size distribution retrieval is possible in areas of the cloud where the dual-wavelength ratio method would fail because the bulk dual-wavelength ratio value is too close to zero.

The ice particles grow rapidly from a maximum size of 0.75 mm to 5 mm while falling less than 500 m and in under 10 minutes. This rapid growth is shown to agree well with theoretical estimates of aggregation, with aggregation efficiency elose

- 15 to lapproximately 0.7, and is inconsistent with other growth processes, e.g. growth by deposition, vapour deposition or riming. The aggregation occurs in the middle of the cloud, and is not present throughout the entire lifetime of the cloud. However, the layer of rapid aggregation is very well defined, at a constant height, where the temperature is  $-15^{\circ}-15^{\circ}C$ , and lasts for at least 20 minutes (approximate horizontal distance: 24 km). Immediately above this layer, the radar Doppler spectra spectrum is bi-modal, which signals the formation of new small ice particles at that height. We suggest that these newly formed particles, at
- 20 approximately  $-15^{\circ}$  C, grow dendritic arms, enabling them to easily interlock and accelerate the aggregation process. The large estimated aggregation efficiency in the studied cloud is between 0.7 and 1, this cloud is consistent with recent laboratory studies for dendrites at this temperature.

A newly developed method for retrieving the ice particle size distribution using the Doppler spectra allows these retrievals in a much larger fraction of the cloud than existing DWR methods. Through quantitative comparison of the Doppler spectra

25 from the three radars we are able to estimate the ice particle size distribution at different heights in the cloud. Comparison of these size distributions with those calculated with more basic radar-derived values and more restrictive assumptions agree very

well; however, the newly developed method allows size distribution retrieval in a larger fraction of the cloud because it allows us to isolate the signal from the larger (non-Rayleigh scattering) particles in the distribution and allows for deviation from the assumed shape of the distribution.

**1 Introduction**

[revised manuscript text omitted]